ns nature communications

# Photocatalytic CO₂ reduction using La-Ni bimetallic sites within a covalent organic framework

Min Zhou[1,7], Zhiqing Wang[1,7], Aohan Mei[1,7], Zifan Yang[1], Wen Chen[1], Siyong Ou[1], Shengyao Wang [2] ✉, Keqiang Chen [1,3] ✉, Peter Reiss [4] ✉, Kun Qi [5], Jingyuan Ma[6] & Yueli Liu [1] ✉

The precise construction of photocatalysts with diatomic sites that simultaneously foster light absorption and catalytic activity is a formidable challenge, as both processes follow distinct pathways. Herein, an electrostatically driven self-assembly approach is used, where phenanthroline is used to synthesize bifunctional LaNi sites within covalent organic framework. The La and Ni site acts as optically and catalytically active center for photocarriers generation and highly selective $CO_2$-to-CO reduction, respectively. Theory calculations and in-situ characterization reveal the directional charge transfer between La-Ni double-atomic sites, leading to decreased reaction energy barriers of *COOH intermediate and enhanced $CO_2$-to-CO conversion. As a result, without any additional photosensitizers, a 15.2 times enhancement of the $CO_2$ reduction rate (605.8 µmol·g⁻¹·h⁻¹) over that of a benchmark covalent organic framework colloid (39.9 µmol·g⁻¹·h⁻¹) and improved CO selectivity (98.2%) are achieved. This work presents a potential strategy for integrating optically and catalytically active centers to enhance photocatalytic $CO_2$ reduction.

Artificial solar-driven $CO_2$ reduction to renewable fuels (e.g., CO[1], HCOOH[2], HCHO[3], CH₃OH[4], and CH₄[5,6]) based on semiconductor-mediated photocatalysis is an intriguing strategy for producing carbon-neutral energy, while also mitigating emerging environmental crises[7,8]. Developing effective photocatalysts with high activity and selectivity has become one of the foremost challenges towards accomplishing this objective[9]. Atomically dispersed catalysts exhibiting maximum atom utilization and unrivaled photoelectric performance are considered the most promising candidates[10,11]. Recently, single-atom catalysts (SACs) with atomically distributed catalytic metal sites have demonstrated impressive catalytic performance in the selective $CO_2$ reduction reaction ($CO_2$RR)[12–14]. However, their limited light absorption capacity and production of numerous intermediates and by-products impede their further implementation. Moreover, the catalytic activity of SACs is limited by the lack of well-defined active sites, inadequate suppression of the competing hydrogen evolution reaction (HER), and weak interaction with the substrate materials[15–17].

Generally, the $CO_2$RR entails three primary processes: photo-absorption, carrier separation, and $CO_2$ reduction[18–21]. Therefore, ideal photocatalysts should possess a sufficient optically active center to generate photoinduced charges as well as an efficient channel to transfer these carriers to a catalytically active center of high selectivity. In contrast to SACs with single catalytic sites, dual-atom catalysts (DACs) with bimetallic centers exhibit tremendous potential for the

[1]State Key Laboratory of Silicate Materials for Architectures, School of Materials Science and Engineering, Wuhan University of Technology, Wuhan 430070, P. R. China. [2]College of Science, Huazhong Agricultural University, Wuhan 430070, P. R. China. [3]Faculty of Materials Science and Chemistry, China University of Geosciences, Wuhan 430070, P. R. China. [4]Univ. Grenoble-Alpes, CEA, CNRS, IRIG/SyMMES, STEP, 38000 Grenoble, France. [5]Institut Européen des Membranes, IEM, UMR 5635, Université Montpellier, ENSCM, CNRS, Montpellier 34000, France. [6]Shanghai Synchrotron Radiation Facility (SSRF), Shanghai Institute of Applied Physics, Chinese Academy of Sciences, Shanghai 200120, P. R. China. [7]These authors contributed equally: Min Zhou, Zhiqing Wang, Aohan Mei. ✉e-mail: wangshengyao@mail.hzau.edu.cn; chenkeqiang@cug.edu.cn; peter.reiss@cea.fr; lylliuwhut@whut.edu.cn

combination of atom-specific characteristics due to the optimized transfer of photogenerated charge carriers and more complex functionalities between adjacent active sites[22,23]. The latter can lead to the enhanced generation of photoinduced charges and improved catalytic activity while maintaining high atom utilization, stability, and properties, which are constrained by the nature of the atomic dispersion[24–29]. As an example, the collaborative coordination of CO at two sets of single-atom sites (Ni and Fe) anchored on nitrogenated carbon has recently been demonstrated to yield an efficient electrocatalyst for the $CO_2RR$, which significantly reduced the reaction barriers for the formation of COOH* and desorption of CO, and resulted in a structural evolution into the CO-adsorbed moiety upon $CO_2$ uptake[30]. Therefore, rationally developed bimetallic site photocatalysts can be used to effectively modulate the selectivity and activity of the $CO_2RR$ by controlling the reaction pathway[31]. However, substantial challenges persist in acquiring a comprehensive understanding of the precise roles of the specific single-atom sites and their interaction occurring at the atomic level during the dynamic photocatalytic $CO_2RR$ process.

Among possible scaffolds for the integration of the dual-atomic sites, crystalline porous network materials such as metal-organic frameworks (MOFs) and covalent organic frameworks (COFs) are renowned for their well-defined porosity, high specific surface area, and predetermined structure[32–36]. These features combined with the confining influence of the ordered pore structure and the abundance of surface binding groups produce a highly favorable environment for the anchoring of metal atoms[37–41]. On the other hand, there are only a few scarce in-depth studies elucidating the critical role of the substrates in the $CO_2RR$ process. From the perspective of DACs, a strong interaction between the catalyst and the substrate is desired to improve the catalytic stability. Furthermore, the interaction between the catalyst and the substrate modulates the processes of carrier separation and transfer and thus has a direct influence on the catalytic performance.

In this paper, we develop an electrostatically driven self-assembly assisted by Phen ligands strategy for incorporating atomically dispersed La-Ni sites (specifically, LaNi-Phen, where Phen= phenanthroline) into conjugated boronate-ester-linked COFs (COF-5 colloid, supplementary Fig. 1)[42,43]. The La-Ni coordination structure is designed to enable efficient production and transfer of photoinduced charges through the optically active (La site) and catalytically active (Ni site) centers. We demonstrate the coordination of the La and Ni sites and unravel the role of the substrate, fostering the effective transfer of the photogenerated carriers. As a result, the optimized LaNi-Phen/COF-5 photocatalyst without any additional photosensitizer exhibits a CO evolution activity of 605.8 μmol·g$^{-1}$ h$^{-1}$ with high selectivity (98.2%), corresponding to a remarkable 15.2-times activity improvement over pristine COF-5 colloid. Experimental characterization data and theoretical calculations confirm that the COF-5 colloid operates as an electron transfer channel through interactions with electrostatically self-assembled La-Phen and Ni-Phen in a mechanism that comprises photoelectron transfer from La-Phen to the COF-5 colloid and subsequent electron injection into Ni-Phen for the $CO_2RR$ process. This work provides a scalable strategy for producing ligand-assisted bimetallic self-assembled COF structures for efficient photocatalytic $CO_2$ reduction.

## Results
### Morphological and structural characterization
La-Ni bimetallic sites are incorporated into COF-5 colloid through a facile electrostatically driven self-assembly assisted by Phen ligands process, in which the La and Ni atoms are captured by B atoms in COF-5 colloid and chelated by the Phen ligands (Fig. 1a), potentially facilitating $CO_2RR$ (Fig. 1b)[44]. Powder X-ray diffraction (PXRD) patterns demonstrate that the crystalline structure of COF-5 colloid remains unaltered throughout the self-assembly process (Supplementary Fig. 2)[45]. The morphology of COF-5, as illustrated in Supplementary Figs. 3 and 4 displays a one-dimensional (1D) nanorods structure with a width distributed around 70–80 nm. Further studies indicate that there is no discernible presence of discrete La-Ni nanoparticles in as-synthesized LaNi-Phen/COF-5 (Fig. 2a), but that the La and Ni ions are

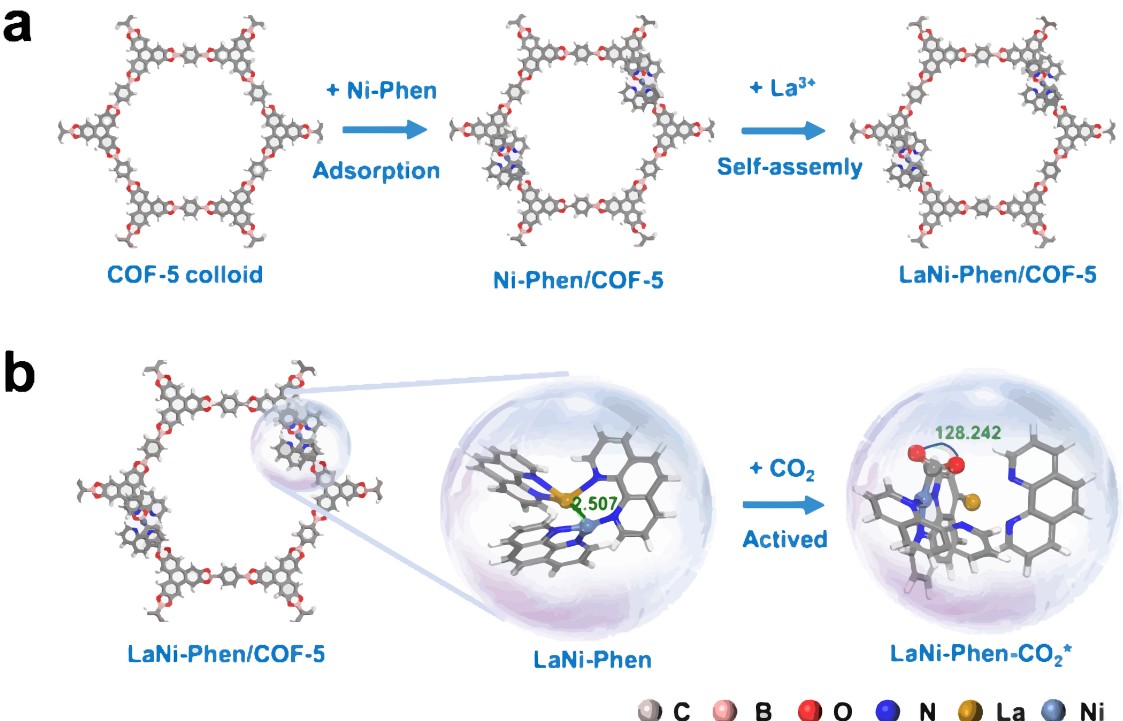

**Fig. 1 | Schematic illustration of the construction of the diatomic sites and $CO_2RR$. a** Self-assembly of LaNi-Phen into COF-5 colloid to create the diatomic site photocatalyst. **b** Schematic diagram of the photocatalytic $CO_2$ reduction in LaNi-Phen/COF-5. Color code: carbon-gray, oxygen-red, boron-pink, hydrogen-white, nitrogen-blue, nickel-light blue, lanthanum-yellow. The same color scheme is applied in Figs. 3 and 6.

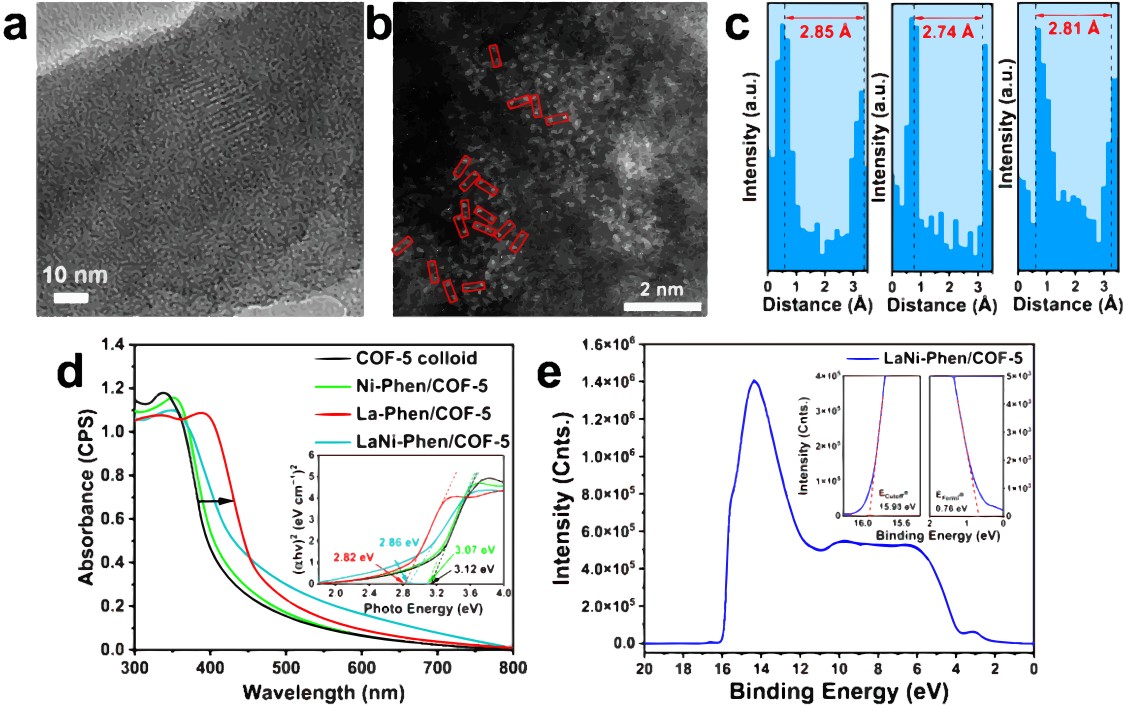

**Fig. 2 | Visualization and spectroscopic characterization of LaNi-Phen/COF-5.**
**a** HR-TEM image, **b** atomic-resolution HAADF-STEM images of LaNi-Phen/COF-5.
**c** measured distance of the representative La-Ni sites in panel **b. d** UV-vis diffuse
reflectance spectra of COF-5 colloid, Ni-Phen/COF-5, La-Phen/COF-5 and LaNi-Phen/
COF-5.(Inset: Tauc plots of LaNi-Phen/COF-5). **e** Full scan UPS spectra of LaNi-Phen/
COF-5 (Inset: Cutoff edge (left) and Fermi edge (right) of LaNi-Phen/COF-5).

uniformly dispersed throughout the COF-5 colloid, with no sign of segregation or aggregation (Supplementary Fig. 5). Importantly, as illustrated in Fig. 2b, c, the atomically dispersed La and Ni ions marked with red box are visible using aberration-corrected high-angle annular dark-field scanning transmission electron microscopy (AC-HAADF-STEM), La-Ni distance of ~2.8 Å can be clear found, indicating the existence of La-Ni double-atomic sites[46]. X-ray photoelectron spectroscopy (XPS) results reveal that the valence states of the Ni and La ions are +2 and +3, respectively (Supplementary Figs. 6–7). In addition, the strong coordination of nitrogen atoms in Phen with Ni and La ions is confirmed by the energetic upshift of the N 1s peaks (Supplementary Note 1 and Supplementary Fig. 8)[47,48]. The FTIR data reveals that Phen can still be observed after the electrostatic self-assembly, as shown in Supplementary Fig. 9 and Supplementary Note 2. The specific elemental contents of La and Ni in LaNi-Phen/COF-5 are calculated to be 2.58 and 1.61 wt%, respectively (Supplementary Fig. 10 and Supplementary Table 1). Notably, the COF-5 colloid in LaNi-Phen/COF-5 comprises a high UV-vis specific surface area, open pore structure, and high $CO_2$ capture capability, ensuring good access to the La-Ni active sites by $CO_2$ molecules in the photocatalytic process (Supplementary Figs. 11–14 and Supplementary Note 3). Moreover, when compared to the host COF-5 colloids, the UV-vis absorption spectrum of LaNi-Phen/ COF-5 exhibits a red-shift indicative of a more favorable energy level alignment for $CO_2$ reduction (Fig. 2d, Supplementary Fig. 15). When comparing to Ni-Phen/COF-5, the modification of COF-5 colloid with La ions alone already leads to a significant red-shift of the absorption onset, implying that the introduction of the rare earth metal La as the optically active center results in enhanced light-harvesting properties and a high degree of electron delocalization in COF-5 colloid. In addition, the bandgap ($E_g$) of COF-5 and LaNi-Phen/COF-5 are estimated to be 3.12 and 2.86 eV by employing Tauc plots (Fig. 2d inset), respectively. The significant reduction of the bandgap facilitates the photogeneration of carriers and improves the light-harvesting properties of LaNi-Phen/COF-5. The Fermi energy level ($E_{Fermi}$) and the valence band maximum (VBM) of LaNi-Phen/COF-5 are determined by

ultraviolet photoelectron spectroscopy (UPS)[49]. Based on the full scan UPS spectra of LaNi-Phen/COF-5 (Fig. 2e), we infer that the cut-off edge energy and Fermi edge energy are 15.93 eV and 0.76 eV, respectively (Fig. 2e inset). Accordingly, the Fermi energy level is calculated as 5.27 eV, while the top position of VB is determined to be −6.03 eV with respect to the vacuum level[50]. Combining with the bandgap ($E_g$ = 2.86 eV) of LaNi-Phen/COF-5, the valence band maximum (VBM) and the conduction band minimum (CBM) of LaNi-Phen/COF-5 are calculated as 1.53 and −1.33 V versus NHE, respectively.

The atomic structure and coordination environment of LaNi-Phen/COF-5 are explored further using X-ray absorption fine structure (XAFS) analysis. The position of the absorption edge of the K-edge X-ray absorption near-edge structure (XANES) spectra is closely related to the coordination environment of the metal atoms (Supplementary Fig. 16a, b). The near-edge absorption energies of LaNi-Phen/COF-5 is located between the metallic nickel and the metal oxide and the white line peak at 8350 eV is higher than that of metallic nickel, indicating that the Ni species is positively charged[51–53]. Furthermore, the dominant Ni-N and La-N peaks in the Fourier transform (FT) k2-weighted R-space extended X-ray absorption fine structure (EXAFS) spectra of LaNi-Phen/COF-5 are near 1.57 and 1.72 Å, respectively (Fig. 3a and b), suggesting the absence of the characteristic peaks of Ni-Ni (≈2.15 Å), La-Ni (≈2.80 Å) and La-La bonding (≈3.94 Å). Consistently, infrared (IR) spectroscopy analysis of the adsorbed CO on La-Phen/COF-5, Ni-Phen/ COF-5, and LaNi-Phen/COF-5 reveals a set of CO absorption bands at 2117 cm⁻¹, which should be assigned to the characteristic frequency for the C-O stretching of the linearly adsorbed CO on La-Ni ionic species (Fig. 3c). Furthermore, no obvious IR peak at 2092 cm⁻¹ corresponding to La-Ni nanoparticles can be observed, confirming the absence of metal nanoparticles or clusters on LaNi-Phen/COF-5 catalysts, which further validates the AC-HAADF-STEM results (Fig. 2c). The intrinsic structure of La and Ni sites is further corroborated with the Fourier-transformed (FT) k3-weighted χ(k)-function of the extended EXAFS spectra in R space. EXAFS fitting analysis at Ni K-edge and La L3-edge show clear small peaks located at 2.90 and 2.88 Å, strongly

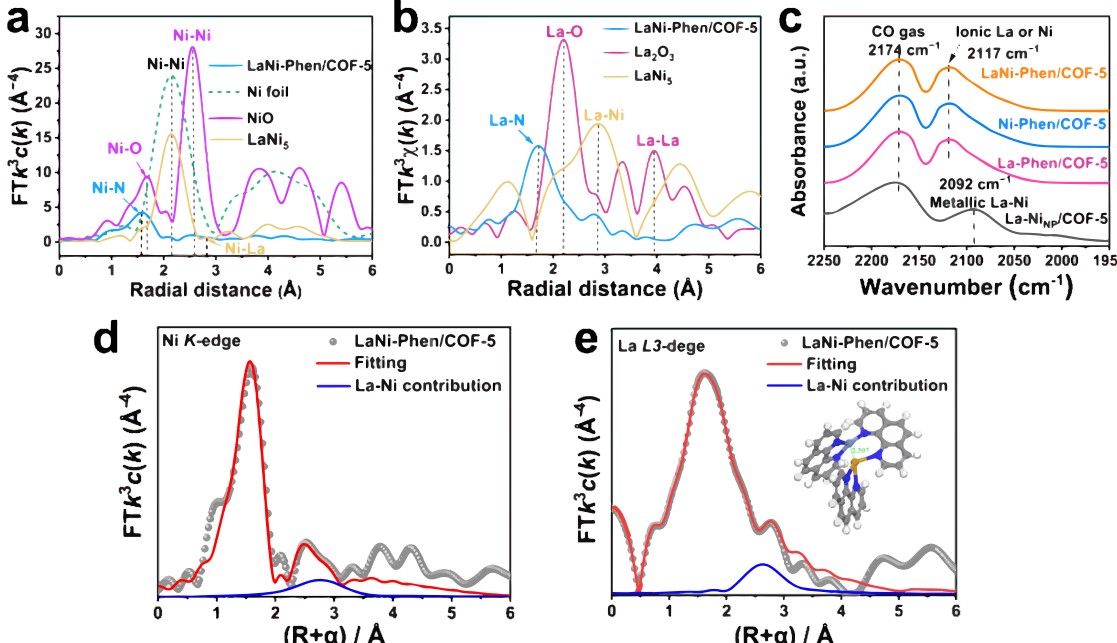

**Fig. 3 | Structural characterization by X-ray absorption spectroscopy. a** Ni K-edge extended X-ray absorption fine structure (R space plots), **b** La L3-edge extended X-ray absorption fine structure (R space plots). **c** DRIFTS spectra of adsorbed CO on La·Phen/COF-5, Ni·Phen/COF-5, LaNi·Phen/COF-5, and a reference material comprising metallic La and Ni. **d** Ni K-edge and **e** La L3-edge EXAFS (points) and curve-fit (line) for LaNi·Phen/COF-5 shown in *R*-space. The data are k³-weighted and not phase-corrected, inset in panel j is the schematic model of the structure of La-N3/Ni-N3.

demonstrating the existence of La-Ni coordination (Fig. 3d, e)[46]. In other words, the atomic dispersion of La and Ni sites as well as direct La-Ni bond are fully confirmed, which further reveals the absence of metal nanoparticles or clusters in LaNi·Phen/COF-5. According to fitting results in Supplementary Table 2, coordination number of La-N and Ni-N could be La-N$_4$/Ni-N$_2$ or La-N$_3$/Ni-N$_3$. Moreover, DFT calculations suggest that the formation energy of La-N$_3$/Ni-N$_3$ structure (−51.760 keV) is lower than that of La-N$_4$/Ni-N$_2$ structure (−51.747 keV), i.e., the La-N$_3$/Ni-N$_3$ structure is preferred (Supplementary Fig. 17). La-N and Ni-N scattering at the first shell over LaNi·Phen/COF-5 with the coordinated structures of La-N$_3$ and Ni-N$_3$ are further confirmed by the EXAFS fitting analyses (Supplementary Figs. 18–21, and Supplementary Table 2), which is consistent with the results of DFT optimization (Supplementary Fig. 22). Free LaNi·Phen is assembled to demonstrate the involvement of the pore confinement effect of COF-5 colloids in facilitating the formation of the La-N$_3$/Ni-N$_3$ coordination structure. The experimental results revealed that the bimetallic La-Ni center eventually self-assembles into a six-coordination structure (La-N$_6$/Ni-N$_6$) in the absence of COF-5 colloid, leading to undesired dual sites because they are fully coordinated in the free state (Supplementary Table 2). This result implies that the pore structure of COFs can significantly promote the precise modulation of the La-Ni diatomic structure, which is at the origin of the formation of bimetallic LaNi·Phen structure in COF-5 colloids.

## CO$_2$ photoreduction activity of LaNi·Phen/COF-5 catalysts

The CO$_2$ reduction reaction is conducted and described in the Methods section to evaluate the catalytic activity of LaNi·Phen/COF-5 under simulated solar irradiation. The CO$_2$RR performance of LaNi·Phen/COF-5 with various La-Ni metal proportions is monitored and optimized systematically (Supplementary Table 3), yielding the highest catalytic activity of 608 μmol·g⁻¹·h⁻¹ (CO) and selectivity of 98.2% (CO over H$_2$). Significantly, the catalytic activity of the COF-5 colloid (39.9 μmol·g⁻¹·h⁻¹), La·Phen/COF-5 (195.4 μmol·g⁻¹·h⁻¹), and Ni·Phen/COF-5 (224.4 μmol·g⁻¹·h⁻¹) is 15.2, 3.1, and 2.7 times lower, respectively (Fig. 4a, b). Moreover, the CO selectivity is substantially higher than

that of COF-5 colloid (70.7%), La·Phen/COF-5 (95.9%), and Ni·Phen/COF-5 (91.2%). We also investigated the CO$_2$RR performance of a physical mixture of La·Phen/COF-5 and Ni·Phen/COF-5 (denoted as mix-LaNi·Phen/COF-5), which exhibits significantly lower catalytic activity with a CO production rate of 115.9 μmol·g⁻¹·h⁻¹. These results unambiguously revealed that La and Ni sites served as predominant catalytic centers and optical centers, respectively, which is facilitated electron transfer and promoted light absorption. Remarkably, the aforementioned activity and selectivity of LaNi·Phen/COF-5 are significantly higher than those of previously reported photocatalysts without unstable noble metal photosensitizers (Supplementary Table 4). In addition, LaNi·Phen/COF-5 exhibits a higher total electron transfer (1,235.0 μmol g⁻¹ h⁻¹) than other catalysts (Supplementary Table 5). As will be shown in the following, the excellent CO$_2$RR performance in this work could be attributed to COF-5's intrinsic structural and electronic characteristics, which facilitates photogenerated carries transfer from La to catalytic center Ni. The foregoing results highlight the crucial role of COF-5 colloid and LaNi·Phen for achieving high performance, and control experiments conclusively confirm that it is an authentic CO$_2$RR process driven by continuous photoexcitation, in which COF-induced confinement effect facilitates the transport of the photogenerated carriers, BIH and H$_2$O operate as electron sacrificial agents and proton sources, respectively (Fig. 4c, Supplementary Figs. 23, 24 and 25). An isotope-labeled carbon dioxide (¹³CO$_2$) photocatalytic reduction experiment based on LaNi·Phen/COF-5 is performed to investigate the origin of the CO$_2$RR products. Three signals generate in the mass spectra (MS), where the total ion chromatographic peak at -7.45 min corresponds to CO (Supplementary Fig. 26). The predominant MS signal at $m/z$ = 29 corresponds to molecular ions of the (¹³CO⁺) peak of ¹³CO, whilst the others (¹³C⁺ at $m/z$ = 13 and O⁺ at $m/z$ = 16) originated from fragments of ¹³CO, demonstrating that CO$_2$ gas is responsible for the formation of the carbon-related products (Fig. 4e). Cycling tests reveal that the CO$_2$RR evolution rate and selectivity exhibit only negligible losses after at least 5 cycles and a total irradiation time of 15 h, demonstrating that the LaNi·Phen/COF-5 catalyst possesses excellent structural robustness

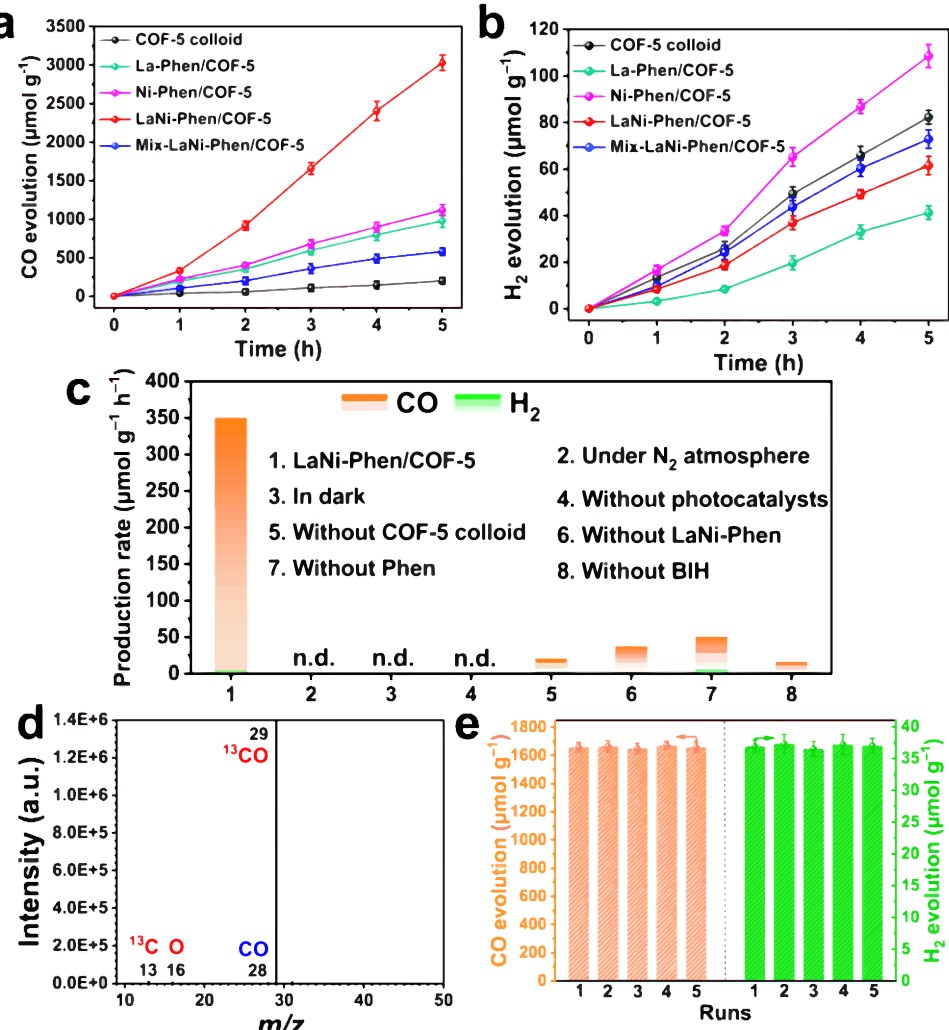

**Fig. 4 | Photocatalytic CO₂ reduction performance. a, b** Time-dependent CO (**a**) and H₂ (**b**) evolution curves under UV-vis light irradiation ($\lambda > 380$ nm) within 5 h using COF-5 colloid (black spheres), La-Phen/COF-5 (green spheres), Ni-Phen/COF-5 (pink spheres), LaNi-Phen/COF-5 (red spheres), and mix-LaNi-Phen/COF-5 (blue spheres). **c** Control experiments of the photocatalytic CO₂ reduction performance over LaNi-Phen/COF-5 under altered conditions (Conditions: 10 mM BIH, 50 mL MeCN, $x$ mL H₂O, 300 W Xe lamp, 298 K). **d** Mass spectra of ¹³CO ($m/z = 29$) produced in the photocatalytic reduction of ¹³CO₂ over LaNi-Phen/COF-5. **e** CO and H₂ production rates in cycling experiments over LaNi-Phen/COF-5.

and durability in the CO₂RR (Fig. 4e). Furthermore, the heterogeneity tests demonstrate the heterogeneous nature of the LaNi-Phen/COF-5 catalysts (Supplementary Fig. 27 and Supplementary Note 4). Moreover, the XRD pattern, FTIR spectra, and XPS spectra of the recovered LaNi-Phen/COF-5 after 5 runs remain unchanged from the as-prepared sample (Supplementary Figs. 28–30). The catalyst's microstructure was retained during the CO₂RR process (Supplementary Figs. 31 and 32), demonstrating that La-Ni-Phen units are strongly maintained in the interior cavities of COF-5 colloid through the confinement effect of the pores. Photoelectrochemical measurements are performed to elucidate the separation and transfer capability of photogenerated carriers over LaNi-Phen/COF-5 (Supplementary Note 5). The steady-state photoluminescence (PL) emission intensity of LaNi-Phen/COF-5 is markedly more attenuated than that of pure COF-5 colloid, owing to improved exciton splitting and charge transfer (Supplementary Fig. 33)[54]. Noteworthy, when COF-5 colloid is only modified by La metal ions, a bathochromic shift (≈26 nm) of the PL peak is observed, suggesting that the modulation of COF-5 colloid's optical properties can be attributed to La ions rather than Ni ions. Time-resolved photoluminescence (TR-PL) measurements illustrate that the carrier lifetimes of the photogenerated charges reduce from 4.75 ns (pure COF-5 colloid) to 0.53 ns (LaNi-Phen/COF-5), demonstrating that the lowering of the recombination of

photogenerated charge carriers is due to the loading of COF-5 colloid with La and Ni ions (Supplementary Fig. 34 and Supplementary Table 6)[55]. In addition, the separation and the transfer of the photogenerated carriers in LaNi-Phen/COF-5 photocatalytic reduction system are investigated by photocurrent-time (I-t) response and electrochemical impedance spectroscopy (EIS) data. The EIS results show the biggest semicircular arc radius of the Nyquist plot in the COF-5 colloid (Supplementary Fig. 35), indicating the highest electron transfer resistance. As the construction of the bimetallic structure favors for charge transfer, LaNi-Phen/COF-5 has the smallest radius compared to La-Phen/COF-5 or Ni-Phen/COF-5, which is further confirmed by the increase in photocurrent density (Supplementary Fig. 36).

## Reaction mechanism of CO₂ photoreduction to CO

The adsorbed CO probe molecule on various catalysts is analyzed using in situ infrared spectroscopy to acquire a thorough understanding of the reaction mechanism of the CO₂ photoreduction. The bands in the range of 1800 to 1900 cm⁻¹ correspond to the C-O stretching vibrations of bridging CO (CO_bridge) adsorbed on the La-Ni dual-atom sites, and the intensity of CO absorption increases with irradiation time (Fig. 5a), whereas no obvious CO_bridge band is observed when using single La or Ni sites on COF-5 (Fig. 5b)[56,57]. These results

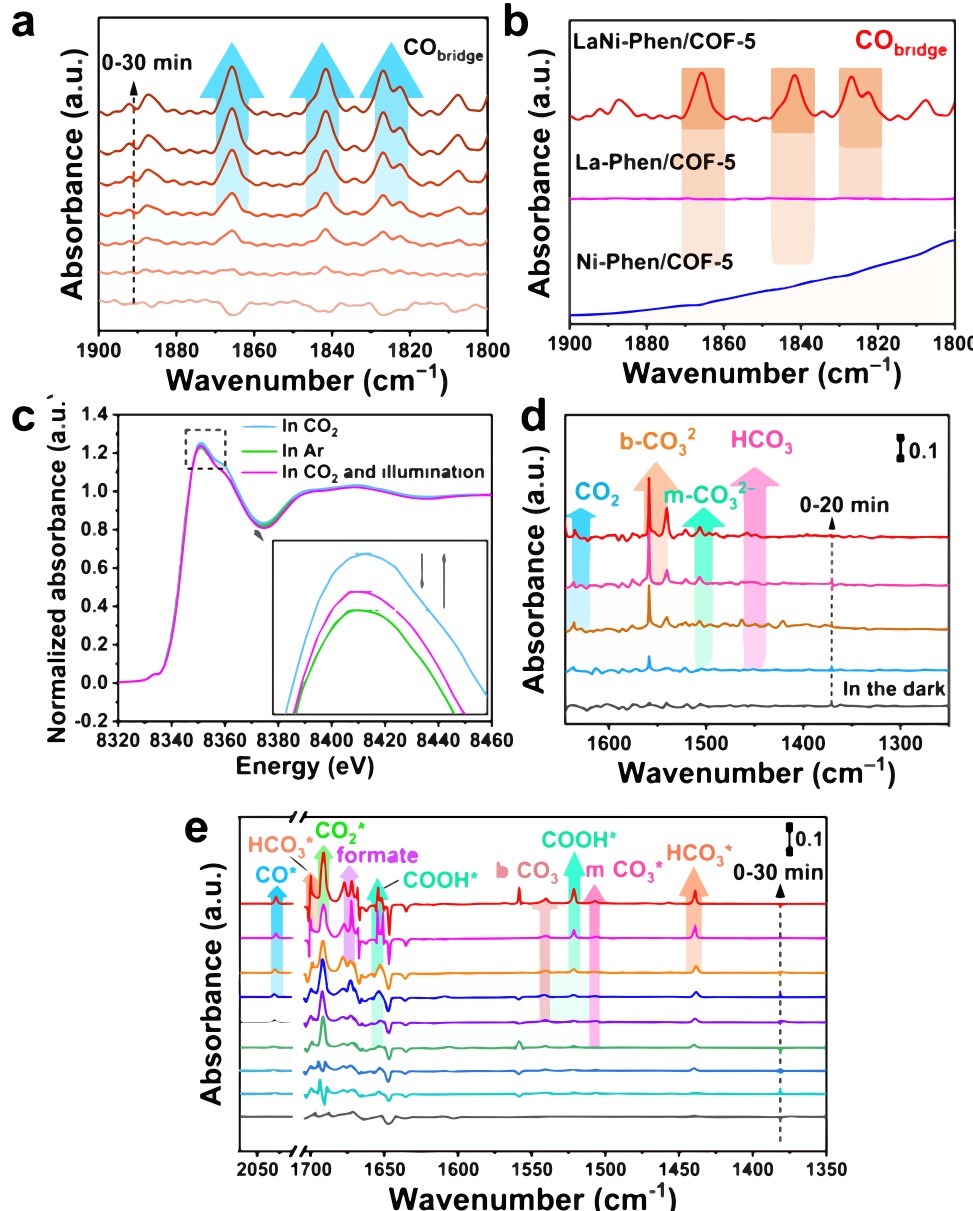

**Fig. 5 | Detection of the reaction mechanism for the photoreduction of $CO_2$ to CO. a** In situ DRIFTS spectra of adsorbed CO at 1800–1900 $cm^{-1}$ for detecting the effective active site over LaNi-Phen/COF-5. **b** DRIFTS spectra of adsorbed CO at 1800–1900 $cm^{-1}$ over La-Phen/COF-5, Ni-Phen/COF-5, and LaNi-Phen/COF-5. **c** Ni K-edge XANES spectra of LaNi-Phen/COF-5 in $CO_2$ photoreduction reduction reaction at room temperature and 1 atm of Ar or $CO_2$-saturated aqueous solution (inset: the enlarged Ni K-edge XANES spectra). **d** In situ DRIFTS spectra of $CO_2$ and $H_2O$ interaction with LaNi-Phen/COF-5 in the dark. **e** In situ DRIFTS spectra at 1350–2070 $cm^{-1}$ for detecting the reaction intermediates in $CO_2$ photoreduction and recording the adsorption and activation of $CO_2$ over LaNi-Phen/COF-5 in the presence of $H_2O$ under subsequent light irradiation.

illustrate unequivocally that only the construction of the La-Ni dual-atom catalyst provides efficient CO adsorption sites, enabling at the same time the efficient electron transfer between the La and Ni sites discussed above as the source of the observed excellent photoelectric performance for the LaNi-Phen/COF-5 catalyst[58]. In situ XAFS measurements are performed to dynamically monitor the oxidation state of active sites to better understand their function on the atomic scale[59]. The Ni K-edge XANES spectra of LaNi-Phen/COF-5 clearly illustrate that the white-line intensity is slightly enhanced in a $CO_2$-saturated acetonitrile solution compared to that in an Ar-saturated acetonitrile solution, revealing the increase of the Ni oxidation state, which is generally ascribed to the spontaneous electron transfer from the active Ni center to the C $2p$ orbital of $CO_2$ during the adsorption and activation of $CO_2$[60]. The intensity of the white line peak is closely

related to the elemental chemistry state, as proven by the calculations of the Ni surface charge, which increases from −0.981 to −0.132 after $CO_2$ adsorption (Supplementary Fig. 37). Interestingly, the white-line intensity of the Ni K-edge XANES spectra reduces slightly under the Xenon lamp irradiation and its position is between that of a $CO_2$-saturated and an Ar-saturated acetonitrile solution, demonstrating the gradual recovery of the active Ni center's oxidation state in the $CO_2RR$ process. Furthermore, an oxidation state variation is observed in the La L3-edge XANES spectra (Supplementary Fig. 38). Summarizing, the in situ XAFS analysis reveals that Ni atoms are the active centers for the $CO_2RR$, while La atoms are not only the optically active center but also the catalytically active center for $CO_2$ adsorption and activation.

The adsorbed surface species and $CO_2$-derived intermediates in the $CO_2RR$ are dynamically monitored using in situ diffuse reflectance

infrared Fourier transform spectroscopy (DRIFTS). The time-resolved spectra of LaNi-Phen/COF-5 after introducing humid $CO_2$ in the dark display the characteristic infrared peaks of active $\cdot CO_2^-$ intermediates ($v_s$(O-C-O): 1634 cm$^{-1}$), bidentate carbonate (b-$CO_3^{2-}$, asymmetric $CO_3$ stretching vibration ($v_{as}$($CO_3$): 1537–1562 cm$^{-1}$), monodentate carbonate (m-$CO_3^{2-}$, $v_s$($CO_3$): 1494–1508 cm$^{-1}$), and bicarbonate $HCO_3^-$ ($\sigma$(CHO): 1439–1462 cm$^{-1}$). Moreover, the $CO_2$ adsorption band ($v_3$($CO_2$)) is indicated by a peak at approximately 3595–3727 cm$^{-1}$, and the intensities represent the $CO_2$ adsorption process in the LaNi-Phen/COF-5 catalysts (Fig. 5d and Supplementary Fig. 39)[61,62]. The subsequent $CO_2RR$ on the surface of LaNi-Phen/COF-5 catalysts relies on an increased $CO_2$ adsorption capacity. As depicted in Fig. 5e, the generation of $CO_2$* at 1691 cm$^{-1}$ with increasing light irradiation time implies activation of $CO_2$ through the route of $CO_2 + e^- \rightarrow CO_2$*, which is primarily ascribed to the facile transfer of the photogenerated electron to $CO_2$ molecules adsorbed on the LaNi-Phen/COF-5 surface[63]. The significant enhancements of the peak intensities at 1540 and 1511 cm$^{-1}$ should be assigned to the COOH* group, which is known as an important intermediate for $CO_2$ reduction[64]. This signifies that the bimetallic LaNi coordination facilitates the generation of abundant *COOH groups, leading to the effectively decreased activation barrier of $CO_2$ transformation. Moreover, the characteristic peak of CO* absorption at 2036 cm$^{-1}$ gradually increases with the prolonging of illumination time, which further reveals the origination of the CO product in the photocatalytic $CO_2RR$ process. Moreover, the $CO_2RR$ process detects the formation of monodentate carbonate (m-$CO_3^{2-}$, $v_s$($CO_3$): 1507 cm$^{-1}$) and bicarbonate $HCO_3^-$ ($\sigma$(CHO): 1439 and 1701 cm$^{-1}$)[65]. Based on these results, $CO_2$*, COOH*, and CO* species should be significant intermediates influencing the photoreduction performance of the LaNi-Phen/COF-5 catalysts[66]. The process of $CO_2RR$ originates from the adsorption of $CO_2$ molecules, followed by the reaction with $H^+$ and the photogenerated electrons to form the intermediate product (*COOH). The *COOH would further lead to the appearing of *CO, and finally CO is desorbed from the catalyst surface. Furthermore, under light irradiation, these intermediates efficiently participate in the $CO_2$ conversion, which is accompanied by a gradual decrease in the intensities of the adsorbed $CO_2$ molecules (Supplementary Fig. 40–41).

Density functional theory (DFT) calculations are carried out to unveil the critical role of LaNi-Phen in the selective photoreduction of $CO_2$ to CO for further investigation of the $CO_2RR$ process over LaNi-Phen/COF-5. The $CO_2$ molecule bends after interacting with LaNi-Phen as illustrated in Fig. 6a, demonstrating that C in activated $CO_2$* interacts with the metallic Ni sites. The calculations demonstrate that the formation energy barrier of COOH* on LaNi-Phen/COF-5 is 0.65 eV, confirming that this process is the rate-limiting step. On the contrary, the formation energy barrier of the CO* intermediate is only 0.04 eV, implying that this process is thermodynamically favorable. In addition, CO desorption is thermodynamically preferred over CHO* formation, with an energy barrier of 0.72 versus 1.02 eV, resulting in a promising photocatalytic performance with a high CO selectivity.

The HOMO-LUMO charge-transfer transitions of LaNi-Phen are also analyzed to further understand the role of atomic La in the $CO_2RR$ enhancement mechanism (Fig. 6b). In particular, we demonstrate that the appropriate electronic characteristics of the La-Ni dual-atomic sites in LaNi-Phen/COF-5 are responsible for providing the electrons for the $CO_2$ photoreduction. Among them, the HOMO energy level in LaNi-Phen is mainly located on Ni-Phen, whereas the LUMO level is on La-Phen, indicating that La-Phen in LaNi-Phen produces the necessary driving force for electron migration from the COF-5 colloid to the bimetallic La-Ni sites. The La atoms act as the optically active center and electron donor, continuously supplying photogenerated electrons to the LaNi-Phen/COF-5 system, while the COF-5 colloid acts as an electron bridge, directing to Ni atoms for the $CO_2$ photoreduction. Moreover, La-Phen exhibits an electrophilic LUMO level after transferring the

photogenerated electron to the COF-5 colloid, leading to the spontaneous replenishment of the excess electrons in Ni-Phen back to La-Phen. This method regulates the product selectivity while enabling closed-loop utilization of the photoinduced charges. The aforementioned result is fully compatible with the proposed efficient charge transfer-induced photocatalysis in LaNi-Phen/COF-5, which combines photoexcited charge-directed transfer with active $CO_2$ adsorption.

The corresponding energy level structure and the hypothetic mechanism for the photocatalytic $CO_2$ reduction with LaNi-Phen/COF-5 are illustrated in Fig. 6c. Notably, the CBM of LaNi-Phen/COF-5 (−1.33 V) is more negative than $E_0$($CO_2$/CO) (−0.53 V) versus NHE. Electrons are photoexcited from the VBM to the CBM of LaNi-Phen/COF-5, which enables the reduction of adsorbed $CO_2$ molecules to CO. Besides, for the oxidation half-reaction, the photoexcited holes in the VB of LaNi-Phen/COF-5 are consumed by the electrons provided by BIH, in which $H_2O$ is oxidized to $O_2$ or $H^+$. Combining the results of in situ characterization and theoretical calculations allows us to corroborate the predicted $CO_2RR$ mechanism of the LaNi-Phen/COF-5 diatomic photocatalyst (Fig. 6d), in which La atoms operate as optically active centers to promote the directional migration of photogenerated carriers and Ni atoms serve as functional catalytically active sites for the adsorption of activated $CO_2$. The design advantage of the DACs system is the appropriate combination of light absorption and catalytic reaction processes on spatially close bimetallic centers supported on COF-5, which can efficiently overcome the longstanding problem of insufficient light absorption capacity for achieving high catalytic efficiency and CO-selectivity.

## Discussion

We present a facile and scalable electrostatically driven self-assembly assisted by Phen ligands to develop a COF-supported photocatalyst comprising La-Ni dual-atom active centers, in which La atoms serve as light-harvesting centers and Ni atoms provide high catalytic activity for the $CO_2$ photoreduction. The presented self-assembly process integrates both the La and Ni atoms into the COF-5 scaffold in single-site forms. The efficient charge transfer between the La-Ni double-atomic sites has been demonstrated using in situ characterizations and theoretical calculations. Specifically, photoelectric characterizations revealed that they accelerate the dynamic behavior of the photogenerated charge carriers, with Ni sites acting as the catalytic centers to promote $CO_2$ photoreduction through COOH* formation and CO* dissociation and La sites favoring photogenerated electron transfer and long-lived charge separation. Therefore, LaNi-Phen/COF-5 exhibits a remarkable CO yield of 605.8 µmol g$^{-1}$ h$^{-1}$ and a high CO selectivity of 98.2% in the absence of any photosensitizer. This work highlights a approach for fabricating dual-atom photocatalysts with well-defined metallic centers inside porous COFs, in which the diatomic sites effectively combine light absorption and catalytic reaction cooperatively to achieve high $CO_2RR$ performance. The straightforward preparation of the described photocatalyst paves the way for the development of related systems with tailored characteristics for the desired catalytic application.

## Methods
### Materials

LaCl$_3$·6H$_2$O (99%), NiCl$_2$·6H$_2$O (99%), and 1,10-Phenanthroline (Phen) were purchased from Aladdin. 2,3,6,7,10,11-Hexahydroxytriphenylene (HHTP) and 1,4-phenylenediboronic acid (PBBA) were provided by Jilin Chinese Academy of Sciences-Yanshen Technology Co., Ltd. as the precursors for COF-5 colloid. Acetonitrile (≥99.5%), mesitylene (98%), and 1,4-dioxane (≥99.5%) were supplied by Aladdin. 1,3-Dimethyl-2-phenyl-2,3-dihydro-1H-benzo[d]imidazole (BIH) was obtained from Bide Pharmatech Ltd. All reagents were of analytical grade and used without further purification.

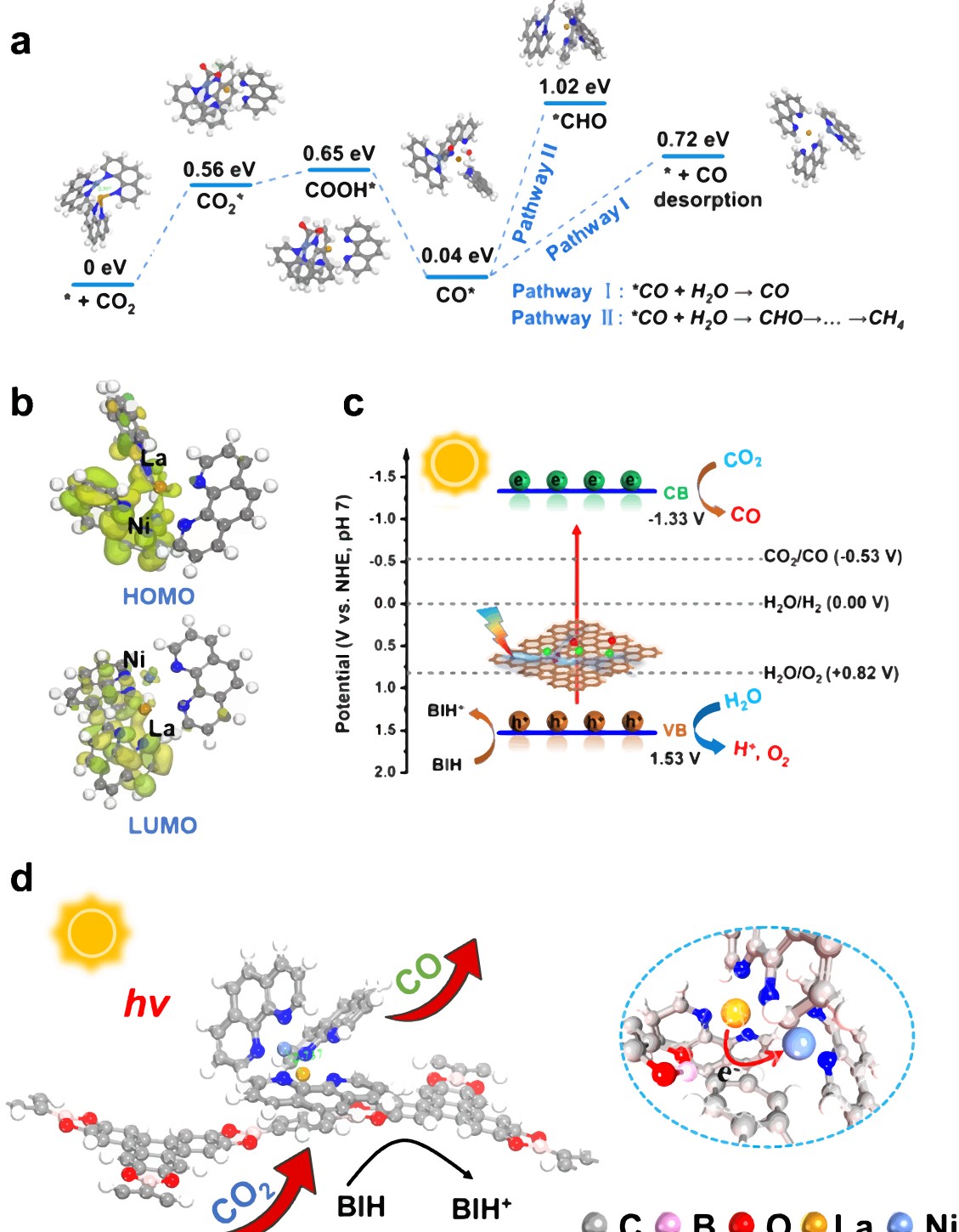

**Fig. 6 | Analysis of the reaction mechanism for the photoreduction of $CO_2$ to CO. a** Reaction pathways for $CO_2$ photoreduction on LaNi·Phen/COF-5 with corresponding geometry structures. **b** HOMO-LUMO charge-transfer transitions for LaNi·Phen. **c** Schematic energy level diagrams and possible reaction mechanism of LaNi·Phen/COF-5 with pH value of 7. **d** Schematic illustration of charge transfer in the photocatalytic $CO_2$ reaction over LaNi·Phen/COF-5.

## Synthesis of COF-5 colloid

Colloidal COF-5 was synthesized by modifying the reported procedures[67,68]. Typically, HHTP (5 mM) and PBBA (7.5 mM) were dissolved in a mixture of acetonitrile: 1,4-dioxane: mesitylene (80:16:4 by volume; 20 mL) and sonicated for 3 min. The solution was subsequently filtered (0.45 μm PTFE) to remove insoluble particles. The solution was then heated to 90 °C without stirring for 24 h under atmospheric pressure. Finally, the resulting COF colloids were

thoroughly washed three times with 10 mL acetone and resuspended with the aforementioned solvent mixture to obtain the final stable COF-5 colloidal suspension.

## Synthesis of LaNi·Phen/COF-5

La and Ni ions were incorporated into the COF-5 colloid through adsorption and chelation with $LaCl_3·6H_2O$, $NiCl_2·6H_2O$, and 1,10-phenanthroline to synthesize LaNi·Phen/COF-5. The COF-5 colloid (3.3 mL,

10 mg) was dispersed in acetonitrile (5 mL) and stirred for half an hour to prepare the precursor solution. Afterward, $NiCl_2 \cdot 6H_2O$ (0.52 mg, 2.2 μmol) was added to the aforementioned solution, which was then stirred for 30 min under an $N_2$ atmosphere. Then, Phen (18 mg, 0.1 mmol) was introduced to the precursor solution and stirred for 1 h to chelate Ni ions and prevent their aggregation. Finally, $LaCl_3 \cdot 6H_2O$ (1.2 mg, 3.3 μmol) was added to the above-mentioned Ni-ion-containing solution and stirred for 12 h at 25 °C under an $N_2$ atmosphere. LaNi-Phen/COF-5 precipitates were collected through centrifugation and then dried for 24 h at 60 °C in a vacuum oven.

### Synthesis of Ni-Phen/COF-5
Following a similar synthetic procedure to that of LaNi-Phen/COF-5, Ni-Phen/COF-5 catalyst was synthesized using 1.3 mg of $NiCl_2 \cdot 6H_2O$ (5.5 μmol), without the addition of any other metal salts.

### Synthesis of La-Phen/COF-5
Following a similar synthetic procedure to that of LaNi-Phen/COF-5, La-Phen/COF-5 was synthesized using 1.9 mg of $LaCl_3 \cdot 6H_2O$ (5.5 μmol).

### Synthesis of Mix-LaNi-Phen/COF-5
The physical mixture of LaNi-Phen/COF-5 (labeled as Mix-LaNi-Phen/COF-5) was prepared by physically mixing 10 mg of La-Phen/COF-5 and 10 mg of Ni-Phen/COF-5.

### Characterization
Powder X-ray diffraction (XRD) patterns of the samples were recorded on a desktop X-ray diffractometer (MiniFlex600/600-C) with Cu Kα radiation ($\lambda = 1.5418$ Å) in the range of $2\theta$ from 2° to 40°. The morphology and microstructure of the samples were examined using High-resolution Transmission Electron Microscope (HRTEM, JEM-2100F, JEOL, Japan) and an Aberration-corrected high-angle annular dark-field scanning transmission electron microscopy (HAADF-STEM, Titan Cube Themis G2 300). The X-ray absorption structures at the Ni K-edge and La L3-edge of the LaNi-Phen/COF-5 were acquired in fluorescence excitation mode using a Lytle detector at the BL14W1-XAFS beamline of the Shanghai Synchrotron Radiation Facility (SSRF). The photon flux of the BL14W1-XAFS beamline is $5 \times 10^{12}$ photons.s$^{-1}$ at 10 keV, with a Si(111) DCM and a beam size of $100 \times 200$ μm$^2$. The $k^3$-weighted $\chi(k)$ data in $k$-space was Fourier-transformed to $R$ space using Hanning windows in the Athena software to separate the EXAFS contributions from different coordination shells. The wavelet transform (WT) of EXAFS spectra was calculated using the Hama Fortran program. The XAFS fitting data was obtained using Artemis software to corroborate the local atomic structure and coordination environment of La and Ni atoms in the LaNi-Phen/COF-5 catalyst. The elemental composition of the samples was evaluated using an Energy Dispersive Spectrometer (EDS). X-ray photoelectron spectroscopy (XPS) measurements were performed on a Thermo ESCALAB 250 spectrometer. The position of the C 1s line at 284.8 eV was utilized to correct all XPS spectra. The Fourier transform infrared (FTIR) spectra of the samples were acquired using a Nicolet 6700 spectrometer. The Brunauer−Emmett-Teller (BET) surface area of the samples at 77 K was determined by $N_2$ adsorption and desorption isotherms using an ASAP 2460 system. The $CO_2$ adsorption capacity of the samples was also tested by ASAP 2460 system at 298 K. The diffuse reflectance spectra (DRS) of the samples were acquired using a UV-vis spectrophotometer (Shimadzu, UV-2550, Japan). The energy band structure of the samples was determined by measuring ultraviolet photoelectron spectra (UPS, ESCALAB 250Xi, Thermo Fisher Scientific, USA). The photogenerated charge carrier separation and lifetime of the samples were recorded using Steady-state Photoluminescence (PL) spectra (LabRam HR, HORIBA Jobin Yvon, France). Time-resolved photoluminescence (TRPL) spectra were recorded on a fluorescence lifetime spectrophotometer (Spirit 1040-8-SHG, Newport, US) at an excitation

wavelength of 365 nm. All of the electrochemical measurements were performed on a CHI-660e workstation (Shanghai Chenhua Instruments Co.), with Pt wire, Ag/AgCl (saturated KCl), and 0.5 M $Na_2SO_4$ solution functioning as the counter electrode, reference electrode, and electrolyte, respectively. Electrochemical impedance spectroscopy (EIS) was measured over the frequencies ranging from 0.1 Hz to 100 kHz with an amplitude of 5 mV. Photocurrent-time (I-t) curves with an interval of 60 s on/off switching were recorded on measured with an applied voltage of 0.2 V vs. Ag/AgCl.

### Photocatalytic activities measurements
10 mg of photocatalyst powder and 10 mM of BIH were mixed in a 50 mL solution containing 48 mL of acetonitrile and 2 mL of $H_2O$ in a Pyrex glass reaction cell coupled to a $CO_2$ reduction system. After the airtight system was completely evacuated using a vacuum pump (no $O_2$ or $N_2$ was detected by gas chromatography), ~80 kPa of high-purity $CO_2$ (99.999%) gas was injected. After the adsorption equilibrium, a 300 W xenon lamp (~100 mW/cm$^2$) was utilized as the light source to irradiate the photocatalytic cell, and the reaction system was kept at 10 °C by cooling water. Gas chromatography (GC-2030, Shimadzu Corp., Japan) equipped with different chromatographic columns was employed to analyze produced $H_2$ and CO.

### Isotope-labeling measurements
The carbon source for the isotope-labeling measurements was $^{13}CO_2$ gas (Isotope purity, 99%, and chemical purity, 99.9%, Tokyo Gas Chemicals Co., Ltd.) instead of $^{12}CO_2$ gas (Chemical purity, 99.999%, Showa Denko Gas Products Co., Ltd.). Typically, 10 mg of photocatalysts, 10 mM of BIH, 48 mL of acetonitrile, and 2 mL of water were loaded into the reaction cell. The $^{13}CO_2$ photoreduction protocol was the same as described above, and the reduction products were further analyzed by gas chromatography-mass spectrometry (JMS-K9, JEOL-GCQMS, Japan and 6890N Network GC system, Agilent Technologies, USA) equipped with two different kinds of columns for detecting the products of $^{13}CO$ (HP-MOLESIEVE, 30 m × 0.32 mm × 25 μm, Agilent Technologies, USA).

### In situ diffuse reflectance infrared Fourier transform spectroscopy (DRIFTS) measurements
In situ DRIFTS spectra were collected using an FT-IR spectrometer (Nicolet iS50 Thermo Scientific, USA) equipped with a mercury cadmium telluride (MCT) detector. Using an ultra-high vacuum pump to eliminate all of the gases in the reaction cell and adsorbed on the catalyst surface. The reaction cell was then filled with humid ultra-pure $CO_2$ gas (99.999%) or CO (99.999%) for $CO_2$ photoreduction or CO adsorption, respectively. Finally, the UV-vis light was turned on, and the in situ DRIFTS data were collected using a difference value of 0 min in light to avoid signals from the organic ligand in the catalysts.

### First-principles-based computational details
We have systematically calculated structural relaxation and electronic characteristics within the framework of Density Functional Theory (DFT) formalism as implemented in Materials Studio (MS). The general gradient approximation (GGA) in the form of a Perdew−Burke−Ernzerhof (PBE) exchange-correlation functional was employed for the self-consistent calculation. The energy cutoff utilized throughout the calculations was set at 700 eV, and the Brillouin zone was sampled using the $2 \times 2 \times 2$ Monkhorst pack for the ionic relaxation of the system. The convergence thresholds for self-consistency and structural relaxation were set at 0.002 Hartree/Å for maximum force, 0.005 Å for maximum displacement, and $1.0 \times 10^{-5}$ Hartree for energy change, respectively.

## Data availability
All data generated and analyzed in this study are included in the paper and its Supplementary Information, and are also available from authors upon request. Source data are provided with this paper.

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

## Acknowledgements

This work was supported by the National Natural Science Foundation of China (12174298), the National Natural Science Foundation of Shenzhen City (No. JCYJ20210324135002007, JCYJ20220530162405012), the Hainan Provincial Joint Project of Sanya Yazhou Bay Science and Technology City (No. 520LH053), Guangdong Basic and Applied Basic Research Foundation (No. 2023A1515010130, 2023A1515030021), the 111 Project (No. B18038), the Fundamental Research Funds for the Central Universities (No. 2021Szvup104), the Natural Science Foundation of Zhejiang Province (No. LZ22A040005), the Natural Science Foundation of Hubei Province (No. 2022CFB654), State Key Laboratory of Advanced Technology for Materials Synthesis and Processing (Wuhan University of Technology) (No. 2022-KF-30), and State Key Laboratory of Silicate Materials for Architectures (Wuhan University of Technology) (No.SYSJJ2022-11). Thanks for the measurements supporting from the Centre for Materials Research and Analysis at Wuhan University of Technology (WUT). We thank the Shanghai Synchrotron Radiation Facility (SSRF) for providing beam time on beamline BL14W1 for the XAS measurements. P.R. acknowledges the support of Agence Nationale de la Recherche through labex ARCANE, ANR-11-LABX-0003-01.

## Author contributions

Y.L., P.R., K.C., S.W., W.C., S.O., and M.Z. conceived and designed the experiments. S.O. and M.Z. carried out the synthesis of materials and photocatalytic test. Z.W. performed the electronic structure calculations. K.Q. and J.M. performed the XENAS test. S.O., M.Z., A.M., and Z.Y. were involved in material test characterization and data analysis. Y.L., P.R., K.C., and S.W. supervised the project. Y.L., P.R., K.C., S.W., W.C., S.O., and M.Z. wrote the paper. All authors discussed the results and commented on the manuscript.

## Competing interests

The authors declare no competing interests.
