## [Peer Review File · Nature Communications]

Photocatalytic CO₂ reduction using La-Ni bimetallic sites within a covalent organic frameworkREVIEWER COMMENTS

Reviewer #1 (Remarks to the Author):

This work looks at the incorporation of a La/Ni bimetallic catalyst for use in the photoreduction of CO₂ using COF-5 as a substrate to prevent deactivation of the catalyst sites through aggregation in order to improve the performance and selectivity of the photocatalytic reaction. The La/Ni catalyst is proven through a process of elimination whereby components are systematically removed to operate synergistically. Much effort went into this work in order to investigate the mechanisms of activity by the measurements of TCSPC, in-situ XAFS, and diffuse reflectance infrared Fourier transform spectroscopy. There are, however, questions I have concerning the nature of the bimetallic catalyst primarily revolving around its characterization and photocatalytic effect. Due to the thoroughness of the work and the importance of its findings, I find this work to be suitable for publication in Nature Communications subject to major revision.

1) To start, I believe that the claimed presence of a self-assembled bimetallic catalyst is unfounded from the evidence given. For instance, little rationale is given for the hypothesis of their self-assembly mechanism. It is stated that they are electrostatically bound to one another, however both metal ions possess a positive charge making it unlikely that they would be coulombically attracted to one another.

2) Second, the EXAFS data provided only utilizes N coordination sites from the FEFF calculation to fit these data and only a single shell is discovered in the wavelet transform plot where a second shell with a Ni/La contribution would be expected if the catalyst were in fact bimetallic at a Ni-La distance of 2.2 Å. It seems more likely that the fitting results using N=2.3 for the Ni-N pass on the Ni K-edge and N=4.2 for the La-N pass on the La L₃-edge could be replaced by N=4 and N=6, respectively to achieve a similar fit; as was done for La(phen)₃ and Ni(phen)₃. If that is not possible other possibilities include the population of remaining coordination sites by chloride ions, especially since chlorine is present in the EDX scan.

3) Finally, it is stated that the increase in photocatalytic performance is, in part, due to the confinement effect (Line 174, Line 226) but in the photocatalytic experiment this effect is not entirely investigated since the concentration of the La(phen)₃ and Ni(phen)₃ in the control study without COF colloid (Figure 3c) could reasonably be much lower than the concentration of the La/Ni confined within the COF colloid. I would suggest attempting to estimate the free volume within 10 mg of the COF colloid and the corresponding concentration of the La/Ni active sites and finally performing a control study with La(phen)₃ and Ni(phen)₃ at those concentrations. By my math the presented hypothetical (phen)₃NiLa would have a molar mass of 738.2 g/mol, going off of the weight % by EDX (which does not include boron) this would be 13% (phen)₃NiLa according to La (2.58% by mass) and 20% (phen)₃NiLa according to Ni (1.61% by mass). Assuming a single cylindrical crystal of 1 hexagon of COF-5 with a molar mass of 2508 A.U./layer (shown below) with a pore diameter of 2.52 nm (Figure S11), and an interlayer distance of 3.5 Å there should be approximately 0.042 cm³ of volume per 10 mg of COF-5. At that volume the concentration of (phen)₃NiLa would be 35.6 mg (phen)₃NiLa per 1 cm³ free volume inside the COF according to La and 59.5 mg (phen)₃NiLa per 1 mL solvent according to the EDX mass % of Ni. If those

concentrations are unattainable due to solubility concerns, it would be suitable to develop a trendline and extrapolate the confinement effect from that data. Notably, these possible concentrations confined within the COF are ~2 orders of magnitude higher than the control experiment performed without COF colloid which appears (the production rate of this control was not reported) to be on par with the catalyst(s) confined within the COF.

In addition, there are some other, more minor things that should be included in the manuscript in order to convince readers of the mechanism of CO₂ photoreduction. These will be listed below:

1) Overall the image quality of the figures needs to be improved. If saving these figures from PowerPoint follow this tutorial to export higher quality (300 dpi or higher) images: How to export high-resolution (high-dpi) slides from PowerPoint - Office | Microsoft Learn

2) Line 119 reads “with a width of 70-80 nm.” I believe this should read “with a width distributed around 70-80 nm.”

3) Line 127 reads (Supplementary Fig. 6, and Supplementary Note 1).” Supplementary Note 1 corresponds to the Supplementary Fig. 7 and so the reference to Supplementary Note 1 should be placed with Supplementary Fig. 7 in Line 129.

4) The interpretation of FTIR data either needs strong literature support or needs to be emphasized less. It is very difficult to tell from the provided spectra whether these bands can be accurately attributed to the formation of metal nitrogen bonds.

5) In Supplementary Table 1 how is Net Error calculated and what is it representing? Also why is boron not appearing in the EDX measurement?

6) The photophysical data for this system is of the utmost importance to this paper since this is a photocatalytic system. In order to convince the readers of the electron transfer from La to Ni the UV-Vis, emission, and the TCSPC data should be brought into the main text. Maybe a good idea to place these next to results of the VBM and CBM relative to CO₂RR and H₂O oxidation from Tauc Plots and UPS.

7) The assignments of the Ni and La XANES data should be supported by either a logical theoretical interpretation or strong literature references. Besides, it seems unlikely that chelation would shift the edge shoulder, as stated on Lines 149-150, since these features are typically the result of ionic charge, or atomic symmetry.

8) The way that Figures 2d and 2e are referred to in the main text (Line 151) should be clearer. These are R space plots.

9) Multiple catalysis trials should be performed and the catalysis results reported with error bars to give the reader a picture of the confidence intervals of the measurements.

10) The physical mixture of La-Phen/COF-5 and Ni-Phen/COF-5 appears to have a deleterious effect on the photocatalytic performance compared to the individual La-Phen/COF-5 and Ni-Phen/COF-5 – why may this be?

11) Again, the in-situ XAFS data needs to be supported with a theoretical interpretation or literature references – the white line intensity decreases, but why is this attributed to an increase in the Ni oxidation state?

12) Some the assignments made in the DRIFTS data are unsupported either by a clear theoretical discussion or literature references. The authors should be clear as to why these bands represent what they claim they do. Some references are provided later in the paragraph about in-situ DRIFTS and it may be that these just need to be referenced earlier.

13) Depending on the interpretation of the EXAFS and catalysis results it may be likely that the DFT calculations need to be revisited.

Overall, I believe the photocatalytic scheme of La acting as photosensitizer and Ni as the active site is sound but the justification for this relies too much on data that is not yet sufficiently interpreted.

Reviewer #2 (Remarks to the Author):

The manuscript tries to understand the role of diatomic sites catalysts towards photocatalytic reduction of CO₂ to selectively produce CO. A ligand mediated method has been used to incorporate La³⁺ as the optical site and Ni²⁺ as the catalytic reduction site. A combined approach of material characterization, DFT calculations, and detailed photocatalytic reactions have been used to prove the efficiency of the system. The detailed investigations have generated a lot of data and the authors have correctly described the observations, but an in depth explanation has not been provided in the main or supplementary text. The manuscript can be subjected to acceptance with the following modifications:

1) The ligand mediated catalyst synthesis is already reported. It might be proper if the authors give due credit to the previous literature (<https://doi.org/10.1038/s41467-019-12510-0>).

2) In supplementary Fig. 13, A) why do the COF-5 have a higher hysteresis loop: Is it because of pore collapse, or kinetic entrapment...? ; B) CO₂ adsorption at a single temperature normally is not the real indicator of the CO₂ sorption ability of a material. Is it possible to do the measurements at 1-2 more different temp. to calculate the heat of adsorption of CO₂?

3) It is unclear as to why in Figure 3c, condition 2, replacing N₂ with CO₂ produced no data. It would provide more clarity, if the reaction conditions are mentioned in the figure caption.

4) Supplementary Fig. 22 and 23, nicely portrays the role of solvent on CO₂ reduction or concentration of water. But no clear explanations have been provided for these phenomena. Also, are these measurements done in absence of sacrificial agent BIH? Why? Can the authors provide more insight in these observations?

5) Supplementary Fig. 31 Transient-state PL spectra, shows the lifetime of photogenerated carrier. According to the authors, how will the synergism work when the lifetime of La phen-COF < La, Ni phen-COF, since this measurement is supposedly taken in absence of CO₂, so there is a possibility that the electrons in HOMO of Ni-phen might get overcrowded, resulting in rapid recombination.

6) Supplementary Fig. 32 Transient photocurrent responses: A) The potential at which the measurements were recorded should be provided. B) The term "Transient photocurrent" might be an over exaggeration since, transient photocurrent (TPC) is measured at timescale of microsecond and under short-circuit conditions. C) For actual comparisons, the photocurrents should be offset at 0 uA/cm².

7) Supplementary Fig. 33: Electrochemical impedance spectra: No explanation as to why the charge transfer restriction reduces. The frequency, amplitude, and potential of measurement of Nyquist plot required to be mentioned.

8) In page 15, a short description of the pathway of the photogenerated carriers for CO₂ reduction has been provided, where the authors claim the formation of a closed loop transfer of electrons from LUMO to HOMO via COF-5, and excess electrons come back to LUMO. A) If such an efficient system can be obtained through synergism of the diatomic catalysts, what is the use/necessity of sacrificial agent? B) What is the CO yield/selectivity without BIH?

Response to the reviewers' comments

Response to Reviewer #1

Comments: This work looks at the incorporation of a La/Ni bimetallic catalyst for use in the photoreduction of CO₂ using COF-5 as a substrate to prevent deactivation of the catalyst sites through aggregation in order to improve the performance and selectivity of the photocatalytic reaction. The La/Ni catalyst is proven through a process of elimination whereby components are systematically removed to operate synergistically. Much effort went into this work in order to investigate the mechanisms of activity by the measurements of TCSPC, in-situ XAFS, and diffuse reflectance infrared Fourier transform spectroscopy. There are, however, questions I have concerning the nature of the bimetallic catalyst primarily revolving around its characterization and photocatalytic effect. Due to the thoroughness of the work and the importance of its findings, I find this work to be suitable for publication in Nature Communications subject to major revision.

Response: Thank you for your constructive comments and hard work, we think that the suggestions are quite useful for us to improve the quality of our manuscript. We have completely addressed your comments point by point and prepared a revised manuscript accordingly. Please find the modifications in the revised manuscript.

Question 1: To start, I believe that the claimed presence of a self-assembled bimetallic catalyst is unfounded from the evidence given. For instance, little rationale is given for the hypothesis of their self-assembly mechanism. It is stated that they are electrostatically bound to one another, however both metal ions possess a positive charge making it unlikely that they would be coulombically attracted to one another.

Response: Thank you very much for your kind advices and hard work as well. We are sorry for the unclear description in the previous version. It is reported that the short-

term interaction (1–5 Å) between the metal atoms in dual-atom catalysts leads to unique electronic structure of the active sites through direct metal-metal bonding or indirect inductive $M_1\text{-N/C-M}_2$ interactions (*Nano Energy*, 2022, 104, 107927). In addition, the distance between the metal atoms of La and Ni is reported to be 2.8 Å (*J. Alloys Compd.*, 2018, 731, 172), which have also been observed in our work (Supplementary Figure 16 in the revised version). That is to say, the self-assemble process of La-Ni bimetals should be attributed to the direct La-Ni bonding and indirect inductive La-N/Ni-N interactions. We have also provided a clearer description to the self-assembly process, please find it in the revised manuscript.

CORRECTIVE ACTIONS:

Lin 5, Abstract:

Herein, a facile electrostatically driven self-assembly assisted by Phen ligands approach is employed to realize a bifunctional architecture of a diatomic LaNi-Phen (Phenanthroline)/covalent organic framework (COF-5) photocatalyst.

Page 6, line 2:

La-Ni bimetallic sites are incorporated into COF-5 colloid through a facile electrostatically driven self-assembly assisted by Phen ligands process, in which the La and Ni atoms are captured by B atoms in COF-5 colloid and chelated by the Phen ligands (Fig. 1a), potentially facilitating CO₂RR (Fig. 1b)⁴⁴.

References:

44. Wang, J., Zhao, C. X., Liu, J. N., Song, Y. W., Huang, J. Q. & Li, B. Q. Dual-atom catalysts for oxygen electrocatalysis. *Nano Energy*, **104**, 107927 (2022).

Fig. 1b:

Fig. 1 (b) Schematic diagram of the photocatalytic CO₂ reduction in LaNi-Phen/COF-5. Color code: carbon-grey, oxygen-red, boron-pink, hydrogen-white, nitrogen-blue, nickel-light blue, lanthanum-yellow. The same color scheme is applied in Fig. 2 and 4.

Question 2: Second, the EXAFS data provided only utilizes N coordination sites from the FEFF calculation to fit these data and only a single shell is discovered in the wavelet transform plot where a second shell with a Ni/La contribution would be expected if the catalyst were in fact bimetallic at a Ni-La distance of 2.2 Å. It seems more likely that the fitting results using N=2.3 for the Ni-N pass on the Ni K-edge and N=4.2 for the La-N pass on the La L3-edge could be replaced by N=4 and N=6, respectively to achieve a similar fit; as was done for La(phen)₃ and Ni(phen)₃. If that is not possible other possibilities include the population of remaining coordination sites by chloride ions, especially since chlorine is present in the EDX scan.

Response: Thank you for your kind suggestions and we think that the suggestions are quite useful for us to improve the quality of our manuscript. First, we do agree with your statement that a second shell with a Ni/La contribution would be expected if the catalyst is in fact bimetallic at a Ni-La distance of 2.2 Å. However, in this work, it is hard to identify the La-Ni shell from the WT data. In contrast, we re-fitted the EXAFS results inspired by your comments, the La-Ni interactions could be clear found in the Fourier transformed (FT) k³-weighted $\chi(k)$ -function of the extended EXAFS spectra in

R space (Fig 2i, 2j, 2.90 and 2.88 Å), which is well in consistent with the previous report (*J. Alloys Compd.*, 2018, 731, 172).

Second, the La-N₆/Ni-N₄ or La-N₆/Ni-N₆ coordination structures should appear in the single-atom metal (La-Phen or Ni-Phen) rather than bimetallic coordination (LaNi-Phen). Besides, the ligand densities in these coordination structures are too high to realize CO₂ adsorption, thus reduce the photocatalytic reduction efficiency. Moreover, we have re-fitted the EXAFS data according to your kind suggestion, it is found that both of La-N₃/Ni-N₃ and La-N₄/Ni-N₂ seems to be reasonable. However, DFT calculations suggest that the formation energy of La-N₃/Ni-N₃ structure (-51.760 keV) is lower than that of La-N₄/Ni-N₂ structure (-51.747 keV), *i.e.*, the La-N₃/Ni-N₃ structure is preferred (Supplementary Fig. 16). In summary, we believe that the coordination structure prefers to be La-N₃/Ni-N₃ instead of La-N₄/Ni-N₂.

CORRECTIVE ACTIONS:

Page 8, Last line:

EXAFS fitting analysis at Ni K-edge and La L3-edge show clear small peaks located at 2.90 and 2.88 Å, strongly demonstrating the existence of La-Ni coordination (Figs. 2i, 2j)⁵³. In other words, the atomic dispersion of La and Ni sites as well as direct La-Ni bond are fully confirmed, which further reveals the absence of metal nanoparticles or clusters in LaNi-Phen/COF-5. According to fitting results in Supplementary Table 2, coordination number of La-N and Ni-N could be La-N₄/Ni-N₂ or La-N₃/Ni-N₃. Moreover, DFT calculations suggest that the formation energy of La-N₃/Ni-N₃ structure (-51.760 keV) is lower than that of La-N₄/Ni-N₂ structure (-51.747 keV), *i.e.*, the La-N₃/Ni-N₃ structure is preferred (Supplementary Fig. 16).

Fig. 2i, 2j:

Fig. 2 (i) Ni K-edge and (j) La L3-edge EXAFS (points) and curve-fit (line) for LaNi-Phen/COF-5 shown in R -space. The data are k^3 -weighted and not phase-corrected, inset in panel j is the schematic model of the structure of La-N3/Ni-N3.

References:

53. Liu, J. J., Zhi, Z., Cheng, H. H., Li, K., Yan, K., Han, X. B., Wang, Y. & Liu, Y. Long-term hydrogen storage performance and structural evolution of LaNi₄Al alloy. *J. Alloys Compd.* **731**, 172-180 (2018).

Supplementary Fig. 16:

Supplementary Fig. 16 Simulated local structure of (a) La-(Phen)₃, (b) Ni-(Phen)₃, (c) La-N₄/Ni-N₂, (d) La-N₃/Ni-N₃.

Page 8, line 10:

Furthermore, the dominant Ni-N and La-N peaks in the Fourier transform (FT) k²-weighted R-space extended X-ray absorption fine structure (EXAFS) spectra of LaNi-Phen/COF-5 are near 1.57 and 1.72 Å, respectively (Fig. 2f and 2g), suggesting the absence of the characteristic peaks of Ni-Ni (≈2.15 Å) and La-La bonding (≈3.94 Å).

Fig. 2f, 2g:

Fig. 2 (f) Ni K-edge extended X-ray absorption fine structure (R space plots), (g) La L3-edge extended X-ray absorption fine structure (R space plots)

Supplementary Table 2:

Supplementary Table 2 EXAFS fitting parameters. EXAFS curve fitting parameters^a for the Ni K-edge and the La L3-edge for different samples.

Sample	Path	N	R / Å	$\sigma^2/\text{Å}^2$	ΔE_0 (eV)	R-factor/%
Ni foil	Ni–Ni ^b	12.0 ^f	2.47	0.006	2.3	0.5
LaNi-Phen/COF-5	Ni–N ^c	3.0	1.82	0.006	2.6	0.6
LaNi-Phen/COF-5	Ni–N ^c	2.3	1.91	0.003	2.3	0.4
La ₂ O ₃	La–O ^d	6.0 ^f	2.57	0.005	4.5	0.4
LaNi-Phen/COF-5	La–N ^e	4.2	2.54	0.007	–3.1	0.2
LaNi-Phen/COF-5	La–N ^e	3.0	2.62	0.008	–3.3	0.3
[La(Phen) ₃] ³⁺	La–N ^e	6.0	2.56	0.004	2.6	0.3
[Ni(Phen) ₃] ²⁺	Ni–N ^e	6.0	2.13	0.005	–2.8	0.5

Question 3: Finally, it is stated that the increase in photocatalytic performance is, in part, due to the confinement effect (Line 174, Line 226) but in the photocatalytic experiment this effect is not entirely investigated since the concentration of the La(phen)₃ and Ni(phen)₃ in the control study without COF colloid (Figure 3c) could reasonably be much lower than the concentration of the La/Ni confined within the COF colloid. I would suggest attempting to estimate the free volume within 10 mg of the COF colloid and the corresponding concentration of the La/Ni active sites and finally performing a control study with La(phen)₃ and Ni(phen)₃ at those concentrations. By my math the presented hypothetical (phen)₃NiLa would have a molar mass of 738.2 g/mol, going off of the weight % by EDX (which does not include boron) this would be 13% (phen)₃NiLa according to La (2.58% by mass) and 20% (phen)₃NiLa according to Ni (1.61% by mass). Assuming a single cylindrical crystal of 1 hexagon of COF-5 with a molar mass of 2508 A.U./layer (shown below) with a pore diameter of 2.52 nm (Figure S11), and an interlayer distance of 3.5 Å there should be approximately 0.042

cm³ of volume per 10 mg of COF-5. At that volume the concentration of (phen)₃NiLa would be 35.6 mg (phen)₃NiLa per 1 cm³ free volume inside the COF according to La and 59.5 mg (phen)₃NiLa per 1 mL solvent according to the EDX mass % of Ni. If those concentrations are unattainable due to solubility concerns, it would be suitable to develop a trendline and extrapolate the confinement effect from that data. Notably, these possible concentrations confined within the COF are ~2 orders of magnitude higher than the control experiment performed without COF colloid which appears (the production rate of this control was not reported) to be on par with the catalyst(s) confined within the COF.

Response: Thank you very much for your comments. We do agree with your comment that the site concentration of the actual photocatalytic active in LaNi-Phen is lower than that of LaNi-Phen/COF-5, and we are sorry for overlooking this important issue. Therefore, a series of experiments are carried out according to your kind suggestion. The photocatalytic performances of LaNi-Phen with various weights are shown in Supplementary Fig. 23 (revised manuscript). It could be found that the CO yields slowly increase with the increasing of the weights of LaNi-Phen within the range of 10-30 mg. However, further increasing the weight of LaNi-Phen results in a negligible variation for the CO yields. These results further confirm the crucial role of the COF-induced confinement effect for the photocatalytic performance.

CORRECTIVE ACTIONS:

Page 11, line 2:

The foregoing results highlight the crucial role of COF-5 colloid and LaNi-Phen for achieving high performance, and control experiments conclusively confirm that it is an authentic CO₂RR process driven by continuous photoexcitation, **in which COF-induced confinement effect facilitates the transport of the photogenerated carriers**, BIH and H₂O operate as electron sacrificial agents and proton sources, respectively (Fig. 3c, Supplementary Fig. 21, 22 and 23).

Supplementary Fig. 23:

Supplementary Fig. 23 Effect of the weight of LaNi-Phen for CO₂ reduction. Effect of the weight of LaNi-Phen for CO₂ reduction in 3 h reaction, proving that confinement effect of COF-5 colloid on LaNi-Phen significantly influences the catalytic activity and product selectivity. The horizontal coordinate is the mass of the added photocatalyst. (Conditions: 10 mM BIH, 50 mL MeCN, x mL H₂O, 300 W Xe lamp, 298 K).

Question 4: Overall the image quality of the Figures needs to be improved. If saving these figures from PowerPoint follow this tutorial to export higher quality (300 dpi or higher) images: How to export high-resolution (high-dpi) slides from PowerPoint - Office | Microsoft Learn

Response: Thank you very much for your kind suggestions. We have changed the images according to your nice comments, please find them in the revised manuscript.

Question 5: Line 119 reads “with a width of 70-80 nm.” I believe this should read “with a width distributed around 70-80 nm.”;

Response: We sincerely thank the reviewer for the careful reading. As suggested by the reviewer, we have corrected the ‘with a width of 70-80 nm’ to be ‘with a width distributed around 70-80 nm’.

CORRECTIVE ACTIONS:

Page 6, line 10:

The morphology of COF-5, as illustrated in Supplementary Fig. 3 and 4 displays a one-dimensional (1D) nanorods structure with a width distributed around 70-80 nm.

Question 6: Line 127 reads (Supplementary Fig. 6, and Supplementary Note 1).” Supplementary Note 1 corresponds to the Supplementary Fig. 7 and so the reference to Supplementary Note 1 should be placed with Supplementary Fig. 7 in Line 129.

Response: We sincerely thank the reviewer for the careful reading. We have checked the position of the Figure notes according to your nice comment, please see the modified manuscript.

CORRECTIVE ACTIONS:

Page 6, line 18:

In addition, the strong coordination of nitrogen atoms in Phen with Ni and La ions is confirmed by the energetic upshift of the N 1s peaks (Supplementary Note 1 and Supplementary Fig. 7)^{45, 46}.

Question 7: The interpretation of FTIR data either needs strong literature support or needs to be emphasized less. It is very difficult to tell from the provided spectra whether these bands can be accurately attributed to the formation of metal nitrogen bonds.

Response: Thank you very much for your suggestion. In our case, we would like to illustrate the strong La-N and Ni-N interactions by the presence of a new N 1s peak located at 399.2 eV (Supplementary Note 1 and Supplementary Fig. 7). The FTIR data are used to support the presence of Phen ligands after the electrostatic self-assembly. Herein, we have modified the description and added several new references according to your kind suggestion.

CORRECTIVE ACTIONS:

Page 6, line 17:

In addition, the strong coordination of nitrogen atoms in Phen with Ni and La ions is confirmed by the energetic upshift of the N 1s peaks (Supplementary Note 1 and Supplementary Fig. 7)^{46,47}. The FTIR data reveals that Phen can still be observed after the electrostatic self-assembly, as depicted in Supplementary Fig. 8 and Supplementary Note 2.

References:

46. Ayan, J., Kaushik, D., Maryam, N., Matthew, A. A., Rahul, B. & Biplab, M. Dual metalation in a two-dimensional covalent organic framework for photocatalytic C-N cross-coupling reactions. *J. Am. Chem. Soc.* **144**, 7822-7833 (2022).

47. Zou, L., Sa, R. J., Zhong, H., Lv, H. W., Wang X.C. & Wang, R.H. Photoelectron transfer mediated by the interfacial electron effects for boosting visible-light-driven CO₂ reduction. *ACS Catal.* **12**, 3550-3557 (2022).

Question 8: In Supplementary Table 1 how is Net Error calculated and what is it representing? Also why is boron not appearing in the EDX measurement?

Response: Thanks for your kind advices. The Net Error is the atomic ratio error, which is directly obtained from the original data. The main reasons for the absence of boron element in the EDX measurement are its relatively small amount, along with its light relative molecular mass and similar characteristic of X-ray values to carbon.

Question 9: The photophysical data for this system is of the utmost importance to this paper since this is a photocatalytic system. In order to convince the readers of the electron transfer from La to Ni the UV-Vis, emission, and the TCSPC data should be brought into the main text. May be a good idea to place these next to results of the VBM and CBM relative to CO₂RR and H₂O oxidation from Tauc Plots and UPS.

Response: Thanks for your kind suggestion. We have modified the photophysical data in the revised manuscript to explain the migration of electrons (Fig. 2, 5).

CORRECTIVE ACTIONS:

Page 7, line 13:

In addition, the bandgap (E_g) of COF-5 and LaNi-Phen/COF-5 are estimated to be 3.12 and 2.86 eV by employing Tauc plots (Fig. 2d inset), respectively. The significant reduction of the bandgap facilitates the photogeneration of carriers and improves the light-harvesting properties of LaNi-Phen/COF-5. The Fermi energy level (E_{Fermi}) and the valence band maximum (VBM) of LaNi-Phen/COF-5 are determined by ultraviolet photoelectron spectroscopy (UPS)⁴⁸. Based on the full scan UPS spectra of LaNi-Phen/COF-5 (Fig. 2e), we infer that the cut-off edge energy and Fermi edge energy are 15.93 eV and 0.76 eV, respectively (Fig. 2e inset). Accordingly, the Fermi energy level is calculated as 5.27 eV, while the top position of VB is determined to be -6.03 eV with respect to the vacuum level⁴⁹. Combining with the bandgap ($E_g = 2.86$ eV) of LaNi-Phen/COF-5, the valence band maximum (VBM) and the conduction band minimum (CBM) of LaNi-Phen/COF-5 are calculated as 1.53 and -1.33 V versus NHE,

respectively.

Figure 2d and 2e:

Figure 2: UV-vis diffuse reflectance spectra of COF-5 colloid, Ni-Phen/COF-5, La-Phen/COF-5 and LaNi-Phen/COF-5. (Inset: Tauc plots of LaNi-Phen/COF-5). (e) Full scan UPS spectra of LaNi-Phen/COF-5. (Inset: Cutoff edge (left) and Fermi edge (right) of LaNi-Phen/COF-5).

Page 17, line 1:

The corresponding energy level structure and the hypothetic mechanism for the photocatalytic CO_2 reduction with LaNi-Phen/COF-5 are illustrated in Fig. 5c. Notably, the CBM of LaNi-Phen/COF-5 (-1.33 V) is more negative than $E_0(\text{CO}_2/\text{CO})$ (-0.53 V) versus NHE. Electrons are photoexcited from the VBM to the CBM of LaNi-Phen/COF-5, which enables the reduction of adsorbed CO_2 molecules to CO. Besides, for the oxidation half-reaction, the photoexcited holes in the VB of LaNi-Phen/COF-5 are consumed by the electrons provided by BIH, in which H_2O is oxidized to O_2 or H^+ .

Figure 5c:

Figure 5: (c) Schematic energy level diagrams and possible reaction mechanism of LaNi-Phen/COF-5 with pH value of 7.

Question 10: The assignments of the Ni and La XANES data should be supported by either a logical theoretical interpretation or strong literature references. Besides, it seems unlikely that chelation would shift the edge shoulder, as stated on Lines 149-150, since these features are typically the result of ionic charge, or atomic symmetry.

Response: Thanks for your kind suggestion. Firstly, we have provided several strong literature references to support the Ni and La XANES data according to your kind suggestion. Secondly, we apologize for our misunderstanding of the edge shoulder shift and we then provided a more proper understanding according to your nice comments. Previous reports suggested that the variation of the near-edge absorption energies in the K-edge XANES spectra is mainly related to the oxidation state of the absorbing atoms, while the intensity of the white line peak is mainly related to the electronic structure and coordination number of the element. Therefore, the variation of the near-side absorption energy and the intensity of the white line peak of LaNi-Phen/COF-5 proves that the Ni species is positively charged.

CORRECTIVE ACTIONS:

Page 8, line 4:

The position of the absorption edge of the K-edge X-ray absorption near-edge structure (XANES) spectra is closely related to the coordination environment of the metal atoms (Supplementary Fig. 15a-b). The near-edge absorption energies of LaNi-Phen/COF-5 is located between the metallic nickel and the metal oxide and the white line peak at 8350 eV is higher than that of metallic nickel, indicating that the Ni species is positively charged⁵⁰⁻⁵².

References:

50. Yan, H. et al. Bottom-up precise synthesis of stable platinum dimers on graphene. *Nat. Commun.* **8**, 1070 (2017).
51. Cheng, Y. et al. Unsaturated edge-anchored Ni single atoms on porous microwave exfoliated graphene oxide for electrochemical CO₂. *Appl. Catal. B Environ.* **243**, 294-303 (2019).
52. Gong, Y. N., Jiao, L., Qian, Y. Y., Pan, C. Y., Zheng, L. R., Cai, X. C., Liu, B., Yu, S. H. & Jiang H. L. Regulating the coordination environment of MOF-templated single-atom nickel electrocatalysts for boosting CO₂ reduction. *Angew. Chem. Int. Ed.* **59**, 2705-2709 (2020).

Question 11: The way that Figures 2d and 2e are referred to in the main text (Line 151) should be clearer. These are R space plots.

Response: Thanks for your kind suggestion. We have modified the descriptions of the cited Figures and added the relevant references based on your nice suggestions.

CORRECTIVE ACTIONS:

Page 8, line 10:

Furthermore, the dominant Ni-N and La-N peaks in the **Fourier transform (FT) k²-weighted R-space extended X-ray absorption fine structure (EXAFS) spectra** of LaNi-Phen/COF-5 are near 1.57 and 1.72 Å, respectively (Fig. 2f and 2g), suggesting the absence of the characteristic peaks of Ni-Ni (≈ 2.15 Å) and La-La bonding (≈ 3.94 Å).

Question 12: Multiple catalysis trials should be performed and the catalysis results reported with error bars to give the reader a picture of the confidence intervals of the measurements.

Response: Thank you for your kind advices. Based on the results of multiple catalysis trials data, the performance graphs (Fig 3a, 3b, and 3e) with error bars are redrawn according to your kind suggestion.

CORRECTIVE ACTIONS:

Figure 3a, 3b, and 3e:

Figure 3: (a-b) Time-dependent CO (a) and H₂ (b) evolution curves under UV-vis light irradiation ($\lambda > 380$ nm) within 5 h using COF-5 colloid (black spheres), La-Phen/COF-5 (green spheres), Ni-Phen/COF-5 (pink spheres) LaNi-Phen/COF-5 (red spheres) and mix-LaNi-Phen/COF-5 (blue spheres). (e) CO and H₂ production rates in cycling experiments over LaNi-Phen/COF-5.

Question 13: The physical mixture of La-Phen/COF-5 and Ni-Phen/COF-5 appears to have a deleterious effect on the photocatalytic performance compared to the individual La-Phen/COF-5 and Ni-Phen/COF-5 – why may this be?

Response: Thanks for the comments. The photocatalytic CO₂ reduction yield is related to the effective photocatalytic active site. The physical mixture of La-Phen/COF-5 and Ni-Phen/COF-5 is a simple stirred mixture with 1:1 mass ratio, as described in *Synthesis of the photocatalysts*. In the *photocatalytic activities measurements*, Mix-LaNi-Phen/COF-5 contains relatively fewer effective photocatalytic active sites compared to pure La-Phen/COF-5 and Ni-Phen/COF-5. More importantly, the bimetallic structure is formed by pure physical mixing rather than electrostatic self-assembly, which does not build an ideal coordination structure. The independent presence of metal atoms has a certain competitive reaction in the process of photocatalytic reactions (*Nat. Catal.* 2021, 4, 719). In other words, more photons will be captured by La atoms, which would reduce the photocatalytic activity of Ni atoms, eventually leading to the decrease of the catalytic efficiency.

Question 14: Again, the in-situ XAFS data needs to be supported with a theoretical interpretation or literature references – the white line intensity decreases, but why is this attributed to an increase in the Ni oxidation state?

Response: Thanks for your comment. In general, the intensity of the white line peak is related to the electronic structure, coordination number and symmetry of the atoms. Taking Ni species as an example, its oxidation state increases during CO₂ reduction, leading to an increasing of the white line intensity. DFT calculation reveals that the oxidation state changes from -0.981 to -0.132 after CO₂ adsorption (Supplementary Fig. 34). When the light source is introduced, the white-line intensity of the Ni K3-edge XANES spectra decreases and locates between that of the CO₂-saturated aqueous solution and Ar, which suggests the gradual recovery of the oxidation state of Ni species during the photocatalytic CO₂RR process, corresponding to the processes of adsorption, activation, and the photocatalytic CO₂RR (*Angew. Chem. Int. Ed.* 2020, 59, 10651-10657).

CORRECTIVE ACTIONS:

Page 13, line 16:

The intensity of the white line peak is closely related to the elemental chemistry state, as proven by the calculations of the Ni surface charge, which increases from -0.981 to -0.132 after CO₂ adsorption (Supplementary Fig. 34).

Supplementary Fig. 34:

Supplementary Fig. 34 Surface charge calculation. (a) Surface charge of Ni atoms and (b) surface charge of Ni atoms after adsorption of CO₂.

Question 15: Some the assignments made in the DRIFTS data are unsupported either by a clear theoretical discussion or literature references. The authors should be clear as to why these bands represent what they claim they do. Some references are provided later in the paragraph about in-situ DRIFTS and it may be that these just need to be referenced earlier.

Response: Thanks for the instructive suggestions. Based on your comments, the DRIFTS data have been further analyzed and discussed, corresponding references have been adjusted, please find them in the revised manuscript.

CORRECTIVE ACTIONS:

Page 14, line 20:

The significant enhancements of the peak intensities at 1540 and 1511 cm⁻¹ should be assigned to the COOH* group, which is known as an important intermediate for CO₂ reduction⁶¹. This signifies that the bimetallic LaNi coordination facilitates the

generation of abundant *COOH groups, leading to the effectively decreased activation barrier of CO₂ transformation. Moreover, the characteristic peak of CO* absorption at 2,036 cm⁻¹ gradually increases with the prolonging of illumination time, which further reveals the origination of the CO product in the photocatalytic CO₂RR process.

Page 15, line 11:

The process of CO₂RR originates from the adsorption of CO₂ molecules, followed by the reaction with H⁺ and the photogenerated electrons to form the intermediate product (*COOH). The *COOH would further lead to the appearing of *CO, and finally CO is desorbed from the catalyst surface.

References:

60. Chang, X. X., Wang, T. & Gong, J. L. CO₂ photo-reduction: insights into CO₂ activation and reaction on surfaces of photocatalysts. *Energy Environ. Sci.* **9**, 2177-2196 (2016).
61. Fu, J. W., Jiang, K. X., Qiu, X. Q., Yu, J. G. & Liu, M. Product selectivity of photocatalytic CO₂ reduction reactions. *Mater. Today* **32**, 222-243 (2019).
62. Sheng, J. P. et al. Identification of halogen-associated active sites on bismuth-based perovskite quantum dots for efficient and selective CO₂-to-CO photoreduction. *ACS Nano* **14**, 13103-13114 (2020).

Question 16: Depending on the interpretation of the EXAFS and catalysis results it may be likely that the DFT calculations need to be revisited.

Response: Thanks for the kind suggestion. We revisit the DFT calculation based on the modified EXAFS data according to your kind suggestion, please find the modification in the revised manuscript.

CORRECTIVE ACTIONS:

Figure 5a and 5b:

Figure 5 (a) Reaction pathways for CO₂ photoreduction on LaNi-Phen/COF-5 with corresponding geometry structures. **(b)** HOMO-LUMO charge-transfer transitions for LaNi-Phen.

Response to Reviewer #2:

Comments: The manuscript tries to understand the role of diatomic sites catalysts towards photocatalytic reduction of CO₂ to selectively produce CO. A ligand mediated method has been used to incorporate La³⁺ as the optical site and Ni²⁺ as the catalytic reduction site. A combined approach of material characterization, DFT calculations, and detailed photocatalytic reactions have been used to prove the efficiency of the system. The detailed investigations have generated a lot of data and the authors have correctly described the observations, but an in depth explanation has not been provided in the main or supplementary text. The manuscript can be subjected to acceptance with the following modifications:

Response: Thank you for your positive comments and hard work, we think that the suggestions are quite useful for us to improve the quality of our manuscript. We have completely addressed your comments point by point and prepared a revised manuscript accordingly. Please find the modifications in the revised manuscript.

Question 1: The ligand mediated catalyst synthesis is already reported. It might be proper if the authors give due credit to the previous literature (<https://doi.org/10.1038/s41467-019-12510-0>).

Response: Thanks for the helpful suggestions. We have cited the references related to the ligand mediated catalyst synthesis according to your kind suggestion, please find it in the revised manuscript.

CORRECTIVE ACTIONS:

References:

42. Yang, H. Z., Shang, L., Zhang, Q. H., Shi, R., Geoffrey, I. N. W., Gu, L. & Zhang, T.R. A universal ligand mediated method for large scale synthesis of transition metal single atom catalysts. *Nat. Commun.* **10**, 4585(2019).

43. Lin, Y. C., Liu, P.Y., Ever, V., Yao, G., Tian, Z. Q., Zhang, L. J. & Chen L. Fabricating single-atom catalysts from chelating metal in open frameworks. *Adv. Mater.* **31**, 1808193 (2019).

Question 2: In supplementary Fig. 13, A) why do the COF-5 have a higher hysteresis loop: Is it because of pore collapse, or kinetic entrapment...? ; B) CO₂ adsorption at a single temperature normally is not the real indicator of the CO₂ sorption ability of a material. Is it possible to do the measurements at 1-2 more different temp. to calculate the heat of adsorption of CO₂?

Response: Thank you for your constructive suggestions. COF-5 is known as a porous material linked by borate ester bonds with a fixed pore size (2.7 nm) and high porosity, which has excellent gas uptake capacities. However, the fixed porosity may bring capillary condensation at higher pressure due to the rapid increased adsorption in COF-5. When the pores in COF-5 are filled, the adsorption isotherm reaches equilibrium, leading to the emergence of capillary condensation and capillary evaporation at different pressures. As a result, hysteresis could be observed during pressure reduction (*J. Am. Chem. Soc.* 2009, 131, 8875; *J. Membr. Sci.* 2019, 572, 588). In addition, the CO₂ adsorption curves of COF-5 and LaNi-Phen/COF-5 at 273 K has also been monitored according to your kind suggestion, which shows a 1.46 mmol g⁻¹ CO₂ uptake (Supplementary Fig. 13a).

CORRECTIVE ACTIONS:

Supplementary Fig. 13:

Supplementary Fig. 13 CO₂ adsorption curves. CO₂ adsorption isotherms of COF-5 colloid and LaNi-Phen/COF-5 at (a) 273 K and (b) 298 K, indicating the excellent CO₂ capture capability of LaNi-Phen/COF-5 as compared to COF-5 colloid.

Question 3: It is unclear as to why in Figure 3c, condition 2, replacing N₂ with CO₂ produced no data. It would provide more clarity, if the reaction conditions are mentioned in the figure caption.

Response: Thank you very much for your kind suggestion. In order to confirm the origination of the photocatalytic products, a series of controlled experiments (such as replacement CO₂ by N₂, removal of light, catalyst, and sacrificial agent, etc.) have been carried out. For condition 2 (replacing N₂ with CO₂), no data, which means no product, is achieved, further confirming that the origination of the photocatalytic products in our work prefers the reduction of CO₂ to the solvent. In addition, we have provided the reaction conditions in the Figure caption according to your kind suggestion, please find them in the revised manuscript.

CORRECTIVE ACTIONS:

Figure 3c:

Figure 3 (c) Control experiments of the photocatalytic CO₂ reduction performance over LaNi-Phen/COF-5 under altered conditions (Conditions: 10 mM BIH, 50 mL MeCN, x mL H₂O, 300 W Xe lamp, 298 K).

Question 4: Supplementary Fig. 22 and 23, nicely portrays the role of solvent on CO₂ reduction or concentration of water. But no clear explanations have been provided for these phenomena. Also, are these measurements done in absence of sacrificial agent BIH? Why? Can the authors provide more insight in these observations?

Response: Thank you very much for your kind suggestions. First, the liquid-phase photocatalytic CO₂ reduction reaction is carried out in a CO₂ saturated solution. The solubility of CO₂ and the dispersibility of catalysts in the solvent will have a significant effect on the catalytic efficiency. Organic solvents such as acetonitrile, ethyl acetate, and DMF are usually chosen to promote the dissolution of CO₂ (*Chem. Soc. Rev.* 2020, 49, 6579; *ACS Energy Lett.* 2021, 6, 3270). In addition, photocatalytic CO₂ reduction reaction involves multiple electrons and protons (*Solar RRL* 2021, 5, 2100154), and H₂O is the most commonly used proton donor. However, a competing reaction, which H₂O is reduced to H₂, may influence the CO₂ reduction process and then the product selectivity. In order to obtain a highly selective and active photocatalytic reaction system, the influence of the proton source, that is the volume of H₂O in the present work, should not be ignored. Second, these measurements are carried out by using 10

mM of BIH as a cavity sacrificial agent, we are sorry for the unclear description in the previous version, and we have modified it in the revised version.

CORRECTIVE ACTIONS:

Supplementary Fig. 21:

Supplementary Fig. 21 Effect of solvents in the CO₂ photoreduction. Effect of solvents in the CO₂ reduction over LaNi-Phen/COF-5 in a 3 h reaction, proving that acetonitrile is the most favorable reaction solvent with a higher activity and selectivity than the other solvents tested. The solution choice in a liquid-phase photocatalytic reaction plays a crucial role in CO₂ solubility and catalyst dispersion, which directly influences the catalytic efficiency. The results suggest that MeCN is the most suitable one. (Conditions: 10 mg photocatalysts, 10 mM BIH, 50 mL solvents, 2 mL H₂O, 300 W Xe lamp, 298 K).

Supplementary Fig. 21:

Supplementary Fig. 22 Effect of the H₂O content in the CO₂ reduction. Effect of the H₂O content in the CO₂ reduction over LaNi-Phen/COF-5 in a 3 h reaction, proving that H₂O significantly influences the catalytic activity and product selectivity. H₂O is the most commonly used proton donor. However, a competing reaction, which H₂O is reduced to H₂, may influence the CO₂ reduction process and then the product selectivity. In order to obtain a highly selective and active photocatalytic reaction system, the influence of the proton source, that is the volume of H₂O in the present work, should not be ignored. The results reveal that 2 mL H₂O in the reaction possesses the best performance. (Conditions: 10 mg photocatalysts, 10 mM BIH, 50 mL MeCN, x mL H₂O, 300 W Xe lamp, 298 K).

Question 5: Supplementary Fig. 31 Transient-state PL spectra, shows the lifetime of photogenerated carrier. According to the authors, how will the synergism work when the lifetime of La phen-COF < La, Ni phen-COF, since this measurement is supposedly taken in absence of CO₂, so there is a possibility that the electrons in HOMO of Ni-phen might get overcrowded, resulting in rapid recombination.

Response: Thank you very much for your kind suggestion. We should emphasize that the lifetime of LaNi-Phen/COF-5 (0.53 ns) is much shorter than that of La-Phen/COF-5 (1.34 ns), the lifetimes had also been summarized in Supplementary Table 6. The

reduced lifetime of LaNi-Phen/COF-5 suggests its enhanced charge carrier separation kinetics.

Question 6: Supplementary Fig. 32 Transient photocurrent responses: A) The potential at which the measurements were recorded should be provided. B) The term “Transient photocurrent” might be an over exaggeration since, transient photocurrent (TPC) is measured at timescale of microsecond and under short-circuit conditions. C) For actual comparisons, the photocurrents should be offset at 0 $\mu\text{A}/\text{cm}^2$.

Response: Thank you very much for your kind suggestions. We are sorry for the unclear description for Supplementary Fig. 33. The photocurrent-time (I-t) curves with an interval of 60 s on/off switching are recorded at an applied voltage of 0.2 V vs. Ag/AgCl. In addition, the term ‘Transient photocurrent’ has been corrected as ‘photocurrent-time (I-t)’, and the photocurrents have been offset at 0 $\mu\text{A}/\text{cm}^2$ according to your kind suggestions. Please find these modifications in the revised manuscript.

CORRECTIVE ACTIONS:

Page 22, Characterization:

Photocurrent-time (I-t) curves with an interval of 60 s on/off switching were recorded on measured with an applied voltage of 0.2 V vs. Ag/AgCl.

Supplementary Fig. 33:

Supplementary Fig. 33 Photocurrent-time (I-t) responses. Photocurrent-time (I-t) responses of COF-5 colloid, Ni-Phen/COF-5, La-Phen/COF-5 and LaNi-Phen/COF-5.

Question 7: Supplementary Fig. 33: Electrochemical impedance spectra: No explanation as to why the charge transfer restriction reduces. The frequency, amplitude, and potential of measurement of Nyquist plot required to be mentioned.

Response: Thank you very much for your kind suggestion. We are sorry for the unclear description for Supplementary Fig. 32. The bimetallic coordination and the porous structure of COF-5 colloids may promote effective transfer of the photogenerated carriers, as a result, the charge transfer restriction is reduced. Moreover, we have added the frequency, amplitude, and potential of measurement of Nyquist plot in the **Characterization** part according to your kind suggestion.

CORRECTIVE ACTIONS:

Page 12, line 13:

In addition, the separation and the transfer of the photogenerated carriers in LaNi-Phen/COF-5 photocatalytic reduction system are investigated by photocurrent-time (I-t) response and electrochemical impedance spectroscopy (EIS) data. The EIS results show the biggest semicircular arc radius of the Nyquist plot in the COF-5 colloid

(Supplementary Fig. 32), indicating the highest electron transfer resistance. As the construction of the bimetallic structure favors for charge transfer, LaNi-Phen/COF-5 has the smallest radius compared to La-Phen/COF-5 or Ni-Phen/COF-5, which is further confirmed by the increase in photocurrent density (Supplementary Fig. 33).

Page 22, Characterization:

Electrochemical impedance spectroscopy (EIS) was measured over the frequencies ranging from 0.1 Hz to 100 kHz with an amplitude of 5 mV.

Question 8: In page 15, a short description of the pathway of the photogenerated carriers for CO₂ reduction has been provided, where the authors claim the formation of a closed loop transfer of electrons from LUMO to HOMO via COF-5, and excess electrons come back to LUMO. A) If such an efficient system can be obtained through synergism of the diatomic catalysts, what is the use/necessity of sacrificial agent? B) What is the CO yield/selectivity without BIH?

Response: Thank you very much for your kind suggestion. The most ideal electron donor for photocatalytic CO₂RR is H₂O, however, the slow kinetics of the water oxidation reaction ($2\text{H}_2\text{O} \rightarrow 4\text{e}^- + 4\text{H}^+ + \text{O}_2$) would limit the reaction rate of the overall photocatalytic CO₂RR (*Chem. Rev.* 2019, 119, 3962). Therefore, in liquid-phase photocatalytic CO₂RR systems, sacrificial agents, such as triethanolamine (TEOA), triethylamine (TEA), BIH (*Appl. Mater. Today* 2021, 23, 101042), are often used as hole depleting agents, Herein, we choose BIH as a sacrificial agent to capture the photogenerated holes and reduce the recombination of the photogenerated electron-hole pairs, which in turn effectively improves the photocatalytic CO₂ reduction efficiency. If no BIH is presented, the yield is as low as 15.74 $\mu\text{mol}\cdot\text{g}^{-1}\cdot\text{h}^{-1}$, which is shown in Fig 3c (condition 8) in the previous version.

Fig. 3 (c) Control experiments of the photocatalytic CO₂ reduction performance over LaNi-Phen/COF-5 under altered conditions (Conditions: 10 mM BIH, 50 mL MeCN, x mL H₂O, 300 W Xe lamp, 298 K).

REVIEWER COMMENTS

Reviewer #1 (Remarks to the Author):

The authors have made improvements in this revised manuscript. However, the proposed structure and some conclusions are not convincing. I would like the authors to further address and clarify the following points.

1. The boron atoms in the COF-5 can be treated as Lewis acid, which usually can't directly bind to metal ions, such as Ni^{2+} and La^{3+} . Is any reference or analog of inorganic molecule reported to support this proposed structure? The author may synthesize the small inorganic complex Ni-La-Phen to support this hypothesis.
2. The synthesis of La/Ni-COF-5 was described in the manuscript, using large amounts of Phen ligand. Is any specific reason to form this proposed bimetallic complex instead of forming $Ni(Phen)_3^{2+}$ and $La(Phen)_2Cl_3$ separately? By the way, the author claims that the band around 2.8 Å in Figure 2i-j is raised by the Ni-La metal bond, but not directly compared to the model complexes, this band could also be raised by outer sphere atoms, such as C atoms from Phen ligands. Anyway, the proposed bimetallic structure is lack of experimental support.
3. COF-5 was reported about two decades ago, and its chemical reactivity has been extensively explored, such as boron-ester group hydrolysis in the water media (highly sensitive to H_2O). The author claimed that the photocatalysis was handled in the mixed solvent of MeCN(48mL) and water (2mL), how did the author overcome this insuperable stability issue?

Reviewer #2 (Remarks to the Author):

The authors have made a very thorough rebuttal to the comments of the referees, who were already quite positive in the first round. They have performed several additional experiments and have finetuned the interpretation of the XANES/EXAFS spectra upon the suggestions of the referees.

I am satisfied with the rebuttal and the modifications made in the manuscript and I recommend publication of the communication.

Response to the reviewers' comments

Response to Reviewer #1

Comments: The authors have made improvements in this revised manuscript. However, the proposed structure and some conclusions are not convincing. I would like the authors to further address and clarify the following points.

Response: Thank you for your constructive comments and hard work, we think that the suggestions are quite useful for us to improve the quality of our manuscript. We have completely addressed your comments point by point and prepared a revised manuscript accordingly. Please find the modifications in the revised manuscript.

1. The boron atoms in the COF-5 can be treated as Lewis acid, which usually can't directly bind to metal ions, such as Ni²⁺ and La³⁺. Is any reference or analog of inorganic molecule reported to support this proposed structure? The author may synthesize the small inorganic complex Ni-La-Phen to support this hypothesis.

Response: Thank you very much for your kind advices and hard work as well. We do agree with your point of view that COF-5, which may be treated as Lewis acid, can't be directly bond to metal ions. In the present manuscript, COF-5 colloids are treated as substrate materials to support Phen through confinement effect, while the Ni²⁺/La³⁺ may bind to Phen through Ni-N/La-N bond. In addition, the Phen-metal interaction has also been widely reported (*Nat. Commun.*, 2018, 9, 221. *Angew. Chem. Int. Ed.* 2018, 57, 7071. *Eur. J. Org. Chem.* 2003, 2003, 1145.).

We have synthesized the small inorganic complex (LaNi-Phen) according to your kind suggestion. The synthesis process is similar with that of LaNi-Phen/COF-5 except the absence of COF-5. Raman and XPS analyses are carried out to characterize this inorganic complex, as shown in Fig. R1 and R2. One can easily find that the band near 301 cm⁻¹ (ascribed to stretching mode (ν) of Ni-N bound, *Nat. Catal.*, 2021, 4, 157.) of LaNi-Phen (Raman spectrum) shows a blue shift compared with that of Ni-Phen. In

addition, three binding energy peaks located at 399.11, 398.66, and 397.93 eV in the deconvoluted N 1s XPS spectrum of LaNi-Phen may be observed, which should be corresponded to pyridinic nitrogen, Ni coordinated pyridinic nitrogen, and La coordinated pyridinic nitrogen species, respectively (*J. Am. Chem. Soc.*, 2022, 144, 7822.). Importantly, comparing with the Ni/La coordinated pyridinic nitrogen peaks in Ni/La-Phen, the positive/negative shifts are observed. These results suggest the formation of LaNi-Phen small inorganic complex.

Fig. R1 Raman spectra. Raman spectra of Phen, La-Phen, Ni-Phen and LaNi-Phen.

Fig. R2 XPS spectra. XPS spectra of La-Phen, Ni-Phen and LaNi-Phen. (a) full spectra, (b) N 1s, (c) Ni 2p and (d) La 3d.

2. The synthesis of La/Ni-COF-5 was described in the manuscript, using large amounts of Phen ligand. Is any specific reason to form this proposed bimetallic complex instead of forming Ni(Phen)₃²⁺ and La(Phen)₂Cl₃ separately? By the way, the author claims that the band around 2.8 Å in Figure 2i-j is raised by the Ni-La metal bond, but not directly compared to the model complexes, this band could also be raised by outer sphere atoms, such as C atoms from Phen ligands. Anyway, the proposed bimetallic structure is lack of experimental support.

Response: Thank you for your kind suggestions and we think that the suggestions are quite useful for us to improve the quality of our manuscript. In general, excessive ligand or substrate are employed to ensure the successful anchoring of atomically dispersed atoms and inhibit their migration and agglomeration (*Joule*, 2018, 2, 1242.). The peak shifts of Raman and XPS spectra indicate the interactions between La and Ni. We have also re-characterized the high-angle annular dark-field scanning transmission electron microscopy (HAADF-STEM) of dual-atoms LaNi to gain further insight into the micro structure of LaNi-Phen/COF-5. The distance of around 2.8 Å indicates the existence of dual-atoms La-Ni. In addition, the additional EXAFS of the LaNi₅ alloy has also been provided according to your kind suggestion, which further confirms the dual-atom

structure of LaNi-Phen/COF-5 (Fig 2f, 2g, and Supplementary Fig. 20 in the revised manuscript).

Page 6, line 13:

Importantly, as illustrated in Fig. 2b-c, the atomically dispersed La and Ni ions marked with red boxes are visible using aberration-corrected high-angle annular dark-field scanning transmission electron microscopy (AC-HAADF-STEM), La-Ni distance of ~ 2.8 Å can be clear found, indicating the existence of La-Ni double-atomic sites.

Fig. 2 (b) atomic-resolution HAADF-STEM images of LaNi-Phen/COF-5. (c) measured distance of the representative La-Ni sites in panel (b).

Page 8, line 10:

Furthermore, the dominant Ni-N and La-N peaks in the Fourier transform (FT) k^2 -weighted R-space extended X-ray absorption fine structure (EXAFS) spectra of LaNi-Phen/COF-5 are near 1.57 and 1.72 Å, respectively (Fig. 2f and 2g), suggesting the absence of the characteristic peaks of Ni-Ni (≈ 2.15 Å), La-Ni (≈ 2.80 Å) and La-La bonding (≈ 3.94 Å).

Fig. 2 (f) Ni K-edge extended X-ray absorption fine structure (R space plots), (g) La L3-edge extended X-ray absorption fine structure (R space plots).

Supplementary Fig. 20 EXAFS fitting curves. Experimental EXAFS data (points) and curve fit (line) at R space (FT magnitude and imaginary component) of (a) Ni sites and (c) La sites for LaNi_5 . The data are k^3 -weighted and not phase-corrected. Experimental EXAFS data (points) and curve fit (line) at k space of (b) Ni sites and (d) La sites for LaNi_5 .

3. COF-5 was reported about two decades ago, and its chemical reactivity has been extensively explored, such as boron-ester group hydrolysis in the water media (highly sensitive to H₂O). The author claimed that the photocatalysis was handled in the mixed solvent of MeCN(48mL) and water (2mL), how did the author overcome this insuperable stability issue?

Response: Thank you very much for your comments. Boronate ester-linker COF-5 may be decomposed when it is exposed to water or moist air, which contributes to the strong electron-deficiency of boron. In the hydrolysis process, one boronate ester bond reacts with one H₂O molecule to break five-membered ring, and then reacts with another H₂O to generate the 2,3,6,7,10,11-Hexahydroxytriphenylene (HHTP) and 1,4-phenylenediboronic acid (PBBA) monomers. However, the π - π interactions with the COF-5 make the energy barrier of the boron center switch from sp^2 to sp^3 hybridization. Therefore, COF-5 has a certain stability in organic solution or low humidity environment (*Adv. Theory Simul.* 2018, 1, 1700015. *J. Am. Chem. Soc.* 2011, 133, 13975-13983.). Additionally, the Phen ligands may be treated as Lewis bases, which may bind with the boron atoms and change its hybridization from sp^2 to sp^3 . This fact may effectively inhibit the interaction between H₂O and the borate bonds, and improve the stability of COF-5. In addition, the XRD patterns shown in Fig. R3(a) further verify the catalytic stability of LaNi-Phen/COF-5. By the way, our another work (under revision consideration by *Angew. Chem. Int. Ed.*) about the ultra-high humidity sensitivity of COF-5 also suggests that its structure may be well maintained under relatively low humidity (RH<43%), as shown in Fig. R3(b).

Fig. R3 (a) XRD patterns of LaNi-Phen/COF-5 treated at different time in the CO₂ atmosphere. (b) XRD patterns of COF-5 film treated with 43% RH for different time;

REVIEWERS' COMMENTS

Reviewer #1 (Remarks to the Author):

The authors have addressed my concerns and I recommend publication in Nature Communications.